# Online Non-convex Learning in Dynamic Environments

**Zhipan Xu**[1], **Lijun Zhang**[1,2,*]

[1]National Key Laboratory for Novel Software Technology, Nanjing University, Nanjing, China
[2]School of Artificial Intelligence, Nanjing University, Nanjing, China
xuzhipan@smail.nju.edu.cn, zhanglj@lamda.nju.edu.cn

## Abstract

This paper considers the problem of online learning with non-convex loss functions in dynamic environments. Recently, Suggala and Netrapalli [2020] demonstrated that follow the perturbed leader (FTPL) can achieve optimal regret for non-convex losses, but their results are limited to static environments. In this research, we examine dynamic environments and choose *dynamic regret* and *adaptive regret* to measure the performance. First, we propose an algorithm named FTPL-D by restarting FTPL periodically and establish $O(T^{\frac{2}{3}}(V_T + 1)^{\frac{1}{3}})$ dynamic regret with the prior knowledge of $V_T$, which is the variation of loss functions. In the case that $V_T$ is unknown, we run multiple FTPL-D with different restarting parameters as experts and use a meta-algorithm to track the best one on the fly. To address the challenge of non-convexity, we utilize randomized sampling in the process of tracking experts. Next, we present a novel algorithm called FTPL-A that dynamically maintains a group of FTPL experts and combines them with an advanced meta-algorithm to obtain $O(\sqrt{\tau \log T})$ adaptive regret for any interval of length $\tau$. Moreover, we demonstrate that FTPL-A also attains an $\tilde{O}(T^{\frac{2}{3}}(V_T + 1)^{\frac{1}{3}})$ dynamic regret bound. Finally, we discuss the application to online constrained meta-learning and conduct experiments to verify the effectiveness of our methods.

## 1 Introduction

Online learning is a powerful model for sequential decision-making tasks, supported by well-established theoretical guarantees [Cesa-Bianchi and Lugosi, 2006, Orabona, 2019]. It can be regarded as a repeated game between a learner and an adversary. In each round $t$, the learner first selects a decision $\mathbf{x}_t \in \mathcal{K}$, where $\mathcal{K}$ is a candidate set. Then a loss function $f_t(\cdot) \colon \mathcal{K} \to \mathbb{R}$ is revealed and the learner suffers a loss $f_t(\mathbf{x}_t)$. The goal of the learner is to minimize the cumulative loss $\sum_{t=1}^{T} f_t(\mathbf{x}_t)$ over all rounds. Traditionally, the performance measure is *static regret* or simply *regret*

$$R(T) = \sum_{t=1}^{T} f_t(\mathbf{x}_t) - \min_{\mathbf{x} \in \mathcal{K}} \sum_{t=1}^{T} f_t(\mathbf{x}), \tag{1}$$

defined as the difference between the cumulative loss of the online learner and that of the optimal decision chosen in hindsight. During the past decades, numerous algorithms and theories have been developed to minimize static regret. An extensively researched setting is online convex optimization (OCO) [Shalev-Shwartz et al., 2012], in which the losses are assumed to be convex and Lipschitz-continuous. For OCO, many efficient methods such as online gradient descent (OGD) attain $O(\sqrt{T})$ regret [Zinkevich, 2003], which is known to be minimax optimal [Abernethy et al., 2008].

---

[*]Lijun Zhang is the corresponding author.

38th Conference on Neural Information Processing Systems (NeurIPS 2024).

On the other hand, when the losses are non-convex, minimizing regret becomes computationally challenging. This is because minimizing regret implies optimization, and general non-convex optimization is known to be NP-hard. In light of this computational barrier, some recent works examine the notion of *local regret* instead [Hazan et al., 2017, Aydore et al., 2019, Hallak et al., 2021, Guan et al., 2023]. However, these researches focus on finding local optima and hence do not guarantee vanishing regret that grows sublinear in $T$. To ensure vanishing regret, another class of studies assumes access to a sampling oracle [Krichene et al., 2015, Yang et al., 2018, Héliou et al., 2020] or an offline optimization oracle [Agarwal et al., 2019, Suggala and Netrapalli, 2020]. Particularly, Suggala and Netrapalli [2020] have demonstrated that for general non-convex and Lipschitz-continuous losses, FTPL can achieve $O(\sqrt{T})$ regret, which matches the optimal result in the convex setting. However, their study solely focuses on static environments, as it uses regret as the only metric to measure the performance. In dynamic environments, the distribution of loss functions may change over time, causing a shift in the optimal decision. In this scenario, the static regret in (1) is not a suitable measure since the comparator is fixed [Zhang, 2020, Cesa-Bianchi and Orabona, 2021]. To overcome this limitation, studies in OCO have introduced new performance metrics: dynamic regret [Zinkevich, 2003, Zhang et al., 2018a] and adaptive regret [Hazan and Seshadhri, 2007, Daniely et al., 2015].

In **dynamic regret** [Zinkevich, 2003], the learner is compared against a sequence of local minimizers:

$$R_D^* = R_D(\mathbf{x}_1^*, \ldots, \mathbf{x}_T^*) = \sum_{t=1}^{T} f_t(\mathbf{x}_t) - \sum_{t=1}^{T} f_t(\mathbf{x}_t^*), \tag{2}$$

where $\mathbf{x}_t^* \in \arg\min_{\mathbf{x} \in \mathcal{K}} f_t(\mathbf{x})$ is a minimizer of $f_t(\mathbf{x})$ over domain $\mathcal{K}$. It is recognized that in the worst-case scenario, achieving sublinear dynamic regret is impossible unless we introduce certain constraints on the comparator sequence or the function sequence [Jadbabaie et al., 2015]. One such example is the functional variation defined below

$$V_T = \sum_{t=2}^{T} \max_{\mathbf{x} \in \mathcal{K}} |f_t(\mathbf{x}) - f_{t-1}(\mathbf{x})|. \tag{3}$$

If the value of $V_T$ is known in advance, Besbes et al. [2015] showed that a restarted OGD achieves $O(T^{\frac{2}{3}}(V_T + 1)^{\frac{1}{3}})$ dynamic regret for convex functions.

**Strongly adaptive regret** [Daniely et al., 2015] is another widely used performance metric in dynamic environments. It is defined as the maximum static regret over any interval of length $\tau$:

$$R_A(T, \tau) = \max_{[s, s+\tau-1] \subseteq [T]} \left\{ \sum_{t=s}^{s+\tau-1} f_t(\mathbf{x}_t) - \min_{\mathbf{x} \in \mathcal{K}} \sum_{t=s}^{s+\tau-1} f_t(\mathbf{x}) \right\}. \tag{4}$$

Given the dynamic nature of environments, where the optimal decisions can vary across intervals, minimizing static regret over any interval of length $\tau$ is essentially competing against changing comparator. For convex functions, the best known result for strongly adaptive regret is $O(\sqrt{\tau \log T})$ [Jun et al., 2017].

In this paper, we consider minimizing dynamic regret and adaptive regret in the non-convex setting. For dynamic regret minimization, we first propose an algorithm named FTPL-D by restarting FTPL periodically and establish an $O(T^{\frac{2}{3}}(V_T + 1)^{\frac{1}{3}})$ dynamic regret bound, which requires the prior knowledge of $V_T$ to set the optimal restarting frequency. To get rid of this limitation, we then propose the second algorithm named FTPL-D+ by running multiple instances of FTPL-D with different restarting frequencies and combining them with a meta-algorithm to track the best one. Given that the loss functions are non-convex, deterministic algorithms cannot achieve a vanishing regret [Suggala and Netrapalli, 2020]. Hence, we choose Hedge [Cesa-Bianchi and Lugosi, 2006] with *randomized sampling* as our meta-algorithm. Correspondingly, in each round, we sample one expert based on their weights instead of computing the weighted average of all experts. We prove that FTPL-D+ enjoys the same order of dynamic regret bound as that of FTPL-D without the prior knowledge of $V_T$.

For adaptive regret minimization, we propose an algorithm named FTPL-A, where we construct a set of intervals dynamically and instantiate an expert of FTPL to minimize the static regret for every interval. To combine these experts whose numbers vary at different time steps, we use AdaNormalHedge [Luo and Schapire, 2015] that can deal with sleeping experts as our meta-algorithm

Table 1: Summary of dynamic regret and strongly adaptive regret bounds. The symbol * indicates that the algorithm requires prior knowledge of $V_T$. The $\tilde{O}(\cdot)$-notation omits logarithmic factors on $T$. Abbreviations: dynamic regret $\to$ D-R, strongly adaptive regret $\to$ SA-R.

| Method | Loss | Metric | Regret Bounds |
|---|---|---|---|
| *Restarted OGD [Besbes et al., 2015] | convex | D-R | $O(T^{\frac{2}{3}}(V_T+1)^{\frac{1}{3}})$ |
| *FTPL-D (ours) | non-convex | D-R | $O(T^{\frac{2}{3}}(V_T+1)^{\frac{1}{3}})$ |
| FTPL-D+ (ours) | non-convex | D-R | $O(T^{\frac{2}{3}}(V_T+1)^{\frac{1}{3}})$ |
| FTPL-A (ours) | non-convex | D-R | $\tilde{O}(T^{\frac{2}{3}}(V_T+1)^{\frac{1}{3}})$ |
| CBCE [Jun et al., 2017] | convex | SA-R | $O(\sqrt{\tau \log T})$ |
| FTPL-A (ours) | non-convex | SA-R | $O(\sqrt{\tau \log T})$ |

and again employ *randomized sampling* in it to select experts. We prove that FTPL-A achieves an $O(\sqrt{\tau \log T})$ adaptive regret bound for any interval of length $\tau$. Besides, we demonstrate that our FTPL-A also obtains an $\tilde{O}(T^{\frac{2}{3}}(V_T+1)^{\frac{1}{3}})$ dynamic regret bound, indicating its effectiveness in minimizing dynamic regret and adaptive regret simultaneously. In brief, our dynamic regret and strongly adaptive regret for non-convex loss functions are on the same order as those for convex functions. We compare our results and previous ones in Table 1.

Moreover, we discuss the application of our methods to online constrained meta-learning [Xu and Zhu, 2023] and conduct experiments. The empirical results demonstrate the effectiveness of our methods in dynamic regret and adaptive regret minimization. We highlight the main contributions of this paper below.

- For dynamic regret minimization, we first propose a novel algorithm named FTPL-D and establish $O(T^{\frac{2}{3}}(V_T+1)^{\frac{1}{3}})$ dynamic regret. To eliminate the dependence on prior knowledge of $V_T$, we then propose FTPL-D+ and provide a dynamic regret bound of the same order.
- For adaptive regret minimization, we develop a novel algorithm named FTPL-A and establish an $O(\sqrt{\tau \log T})$ strongly adaptive regret bound. Moreover, we prove that FTPL-A also ensures an $\tilde{O}(T^{\frac{2}{3}}(V_T+1)^{\frac{1}{3}})$ dynamic regret bound.
- We discuss the application to online constrained meta-learning and conduct experiments to verify the effectiveness of our methods.

## 2   Related Work

In this section, we give a brief introduction to previous works in OCO and online non-convex learning. More related works are reviewed in Appendix B.

**Online Convex Optimization**   The existing works in OCO mostly focus on static regret. For instance, OGD [Zinkevich, 2003] and follow the regularized leader [Hazan et al., 2016] both achieve an $O(\sqrt{T})$ regret bound. The $O(\sqrt{T})$ regret bound for Lipschitz-convex functions is known to be minimax optimal [Abernethy et al., 2008]. To cope with dynamic environments, Zinkevich [2003] introduced the dynamic regret in (2), and there are numerous studies dedicated to the worst-case scenario because of its mathematical tractability [Besbes et al., 2015, Jadbabaie et al., 2015, Mokhtari et al., 2016, Yang et al., 2016, Zhang et al., 2017, Baby and Wang, 2019, Zhao and Zhang, 2021, Wan et al., 2023]. As mentioned earlier, dynamic regret is often bounded in terms of certain regularities of the comparator sequence or the function sequence. In particular, Besbes et al. [2015] proposed the functional variation in (3) to evaluate the movement of loss functions. They equipped restarted OGD with an $O(T^{\frac{2}{3}}(V_T+1)^{\frac{1}{3}})$ dynamic regret bound, but require to know $V_T$ beforehand.

Besides dynamic regret, another metric in dynamic environments is adaptive regret. Adaptive regret has been examined under the setting of prediction with expert advice (PEA) [Littlestone and Warmuth, 1994, Freund et al., 1997, Gyorgy et al., 2012, Adamskiy et al., 2012, Luo and Schapire, 2015] and OCO [Hazan and Seshadhri, 2007, Daniely et al., 2015, Jun et al., 2017, Zhang et al., 2019, 2021, Yang et al., 2024]. In the following, we focus on the latter one. To minimize (4), Daniely et al. [2015] developed a meta-algorithm called strongly adaptive online learner and used it to design

two-layer structured online algorithms, which enjoy an $O(\sqrt{\tau} \log T)$ strongly adaptive regret bound for convex functions. Later, Jun et al. [2017] proposed a novel meta-algorithm and improved the strongly adaptive regret bound to $O(\sqrt{\tau \log T})$.

**Online Non-convex Learning** To avoid the NP-hardness of non-convex optimization, Hazan et al. [2017] proposed a computational tractable notion of local regret and developed algorithms that attain the optimal local regret bound efficiently. Hallak et al. [2021] extended the local regret minimization framework to non-smooth settings. Guan et al. [2023] further examined the cases where a limited number of gradient oracles or value oracles are available. Although these techniques can efficiently minimize local regret, they do not guarantee to find the global optima and achieve vanishing regret.

Another line of work still focuses on the notion of static regret, but assumes access to a sampling oracle or an offline optimization oracle. Assuming access to a sampling oracle, Krichene et al. [2015] proved that the Hedge algorithm is capable of achieving $O(\sqrt{T \log T})$ regret over a specific feasible set. Yang et al. [2018] later improved the regret bound to $O(\sqrt{T})$ by partitioning the feasible set with a layered structure and using a novel weighting method. Besides, Héliou et al. [2020] examined the dual averaging (DA) algorithm with an imperfect value-feedback model of the loss function, which also attained an $O(\sqrt{T})$ regret bound. However, these algorithms rely on a sampling oracle on a continuum that is computationally *intractable*.

Under the hypothesis that the algorithm has access to an offline optimization oracle, Agarwal et al. [2019] showed that FTPL achieves $O(T^{\frac{2}{3}})$ regret for general non-convex loss functions with Lipschitz continuousity. In the same setting, Suggala and Netrapalli [2020] improved the regret bound to $O(\sqrt{T})$. It should be noted that assuming access to an offline optimization oracle is reasonable since some simple algorithms such as stochastic gradient descent, are able to find approximate global optima quickly, even for non-convex objective functions.

By contrast, the studies on dynamic environments are limited. Aydore et al. [2019] introduced a variant of local regret for dynamic environments and proposed a novel algorithm to minimize it. Héliou et al. [2020] obtained $O(T^{\frac{2}{3}}(V_T + 1)^{\frac{1}{3}})$ dynamic regret with their imperfect feedback method. However, it needs the prior knowledge of $V_T$ to choose the optimal stepsize in DA and still relies on a computationally intractable sampling oracle.

## 3 Preliminaries

In this section, we introduce the problem setting and FTPL [Suggala and Netrapalli, 2020].

**Assumption 1.** *The feasible set $\mathcal{K}$ is bounded and has $\ell_\infty$-diameter at most $D$, i.e., for all $\mathbf{x}, \mathbf{y} \in \mathcal{K}$, $\|\mathbf{x} - \mathbf{y}\|_\infty \le D$.*

**Assumption 2.** *The sequence of loss functions $f_t(\cdot)$ are $L$-Lipschitz with respect to $\ell_1$-norm, i.e., for all $\mathbf{x}, \mathbf{y} \in \mathcal{K}$, $|f_t(\mathbf{x}) - f_t(\mathbf{y})| \le L \|\mathbf{x} - \mathbf{y}\|_1$.*

We proceed to introduce FTPL in Algorithm 1, which is the subroutine of our algorithms. FTPL relies on the offline optimization oracle below [Suggala and Netrapalli, 2020].

**Definition 1.** An **offline optimization oracle** takes input as a function $f(\cdot) : \mathcal{K} \to \mathbb{R}$ and a $d$-dimensional vector $\sigma$, and returns an approximate minimizer of $\mathbf{x} \mapsto f(\mathbf{x}) - \langle \sigma, \mathbf{x} \rangle$. An optimization oracle is called "$(\alpha, \beta)$-approximate optimization oracle" if it returns $\mathbf{x}^* \in \mathcal{K}$ such that

$$f(\mathbf{x}^*) - \langle \sigma, \mathbf{x}^* \rangle \le \inf_{\mathbf{x} \in \mathcal{K}} [f(\mathbf{x}) - \langle \sigma, \mathbf{x} \rangle] + (\alpha + \beta \|\sigma\|_1).$$

We denote such an optimization oracle with $\mathcal{O}_{\alpha, \beta}(f_i - \sigma)$.

Given the access to an $(\alpha, \beta)$-approximate optimization oracle, the main idea of FTPL is to add a small perturbation to the cumulative loss and follow the "perturbed" leader:

$$\mathbf{x}_t = \mathcal{O}_{\alpha, \beta} \left( \sum_{i=1}^{t-1} f_i - \langle \sigma_t, \cdot \rangle \right), \tag{5}$$

where $\sigma_t \in \mathbb{R}^d$ is a random perturbation such that $\sigma_{t,j}$, the $j$-th coordinate of $\sigma_t$, is sampled from the exponential distribution with parameter $\eta$, that is

$$\{\sigma_{t,j}\}_{j=1}^d \overset{i.i.d}{\sim} \mathrm{Exp}(\eta). \tag{6}$$

---
**Algorithm 1** Follow the Perturbed Leader (FTPL)
---
1: **Input:** feasible set $\mathcal{K}$, approximation optimization oracle $\mathcal{O}_{\alpha,\beta}$, parameter of exponential distribution $\eta$
2: **for** $t = 1$ **to** $T$ **do**
3:     Generate random vector $\sigma_t$ by (6)
4:     Predict $\mathbf{x}_t$ according to (5)
5:     Observe loss function $f_t(\cdot)$
6: **end for**
---

---
**Algorithm 2** FTPL-D
---
1: **Input:** feasible set $\mathcal{K}$, length of interval $\gamma$, approximation optimization oracle $\mathcal{O}_{\alpha,\beta}$, parameter of exponential distribution $\eta$
2: **for** $t = 1$ **to** $T$ **do**
3:     Compute $s_\gamma = \lfloor (t-1)/\gamma \rfloor * \gamma + 1$
4:     Generate random vector $\sigma_t$ by (6)
5:     Predict $\mathbf{x}_t$ according to (7)
6:     Observe loss function $f_t(\cdot)$
7: **end for**
---

***Remark***. We choose FTPL as our subroutine due to its simplicity and reasonable assumption of the optimization oracle. However, it is worth noting that our methods are not limited to FTPL alone. We can choose any algorithm that has a static regret guarantee in online non-convex learning and extend it to dynamic environments.

## 4 Online Non-convex Learning with Dynamic Regret

In this section, we propose our first algorithm named FTPL-D based on FTPL and establish a dynamic regret bound with the prior knowledge of $V_T$. After that, we propose the second algorithm named FTPL-D+ which is equipped with the same bound without the prior knowledge of $V_T$.

### 4.1 Follow the Perturbed Leader with Dynamic Regret

Let dynamic regret defined in (2) be the performance measure. Our goal is to bound the dynamic regret by the functional variation $V_T$.

Following the work of Besbes et al. [2015], we apply the restarting strategy to FTPL. The key idea is to partition the time horizon $T$ into consecutive intervals, where the length of each interval is controlled by a parameter $\gamma$, and then restart FTPL at the beginning of each interval. Our algorithm, which we call follow the perturbed leader with dynamic regret (FTPL-D), is described in Algorithm 2. At each time step $t$, we mark the beginning of the current interval as $s_\gamma$ in Step 3. Then FTPL-D follows the perturbed leader within the current interval, which spans from $s_\gamma$ to $t$. Specifically,

$$\mathbf{x}_t = \mathcal{O}_{\alpha,\beta} \left( \sum_{i=s_\gamma}^{t-1} f_i - \langle \sigma_t, \cdot \rangle \right). \tag{7}$$

The following theorem presents the dynamic regret of Algorithm 2 for general non-convex functions.

**Theorem 1.** *Under Assumptions 1 and 2, and setting $\eta = 1/\sqrt{d\gamma}$, Algorithm 2 ensures*

$$\mathbb{E}\left[R_D^*\right] \leq \frac{2c(\alpha, \beta, \gamma)T}{\sqrt{\gamma}} + 2\gamma V_T,$$

*where*

$$c(\alpha, \beta, \gamma) = 125DL^2 d^{\frac{3}{2}} + \frac{(21\beta\gamma + D)d^{\frac{3}{2}}}{20} + \frac{21\alpha\sqrt{\gamma}}{20} + 2dL\beta\gamma. \tag{8}$$

*If the value of $V_T$ is known, by choosing $\gamma = \min\left\{ \left\lfloor \left(\frac{T}{V_T}\right)^{\frac{2}{3}} \right\rfloor, T \right\}$, we have*

$$\mathbb{E}\left[R_D^*\right] \leq O\left( (1 + \alpha\sqrt{T} + \beta T)T^{\frac{2}{3}}(V_T + 1)^{\frac{1}{3}} \right).$$

---

**Algorithm 3** FTPL-D+

---

1: **Input:** feasible set $\mathcal{K}$, $\mathcal{H} = \{\gamma_1, ..., \gamma_N\}$, step size $\rho$
2: Activate a set of experts $\{E_i \mid \gamma_i \in \mathcal{H}\}$ by invoking Algorithm 2 for each $\gamma_i \in \mathcal{H}$
3: For each expert $E_i$, set $w_1^i = 1/N$ for $i \in [N]$
4: **for** $t = 1$ **to** $T$ **do**
5:     Receive $\mathbf{x}_t^i$ from each expert $E_i$
6:     Draw $\mathbf{x}_t$ according to $P\left(\mathbf{x}_t = \mathbf{x}_t^i\right) = w_t^i$
7:     Output $\mathbf{x}_t$ and observe loss function $f_t(\cdot)$
8:     Update the weight of experts according to (9)
9:     Send loss function $f_t(\cdot)$ to each expert $E_i$
10: **end for**

---

***Remark.*** Theorem 1 demonstrates that for general non-convex functions, when $\alpha = O(1/\sqrt{T})$ and $\beta = O(1/T)$, which mirrors the settings used by Suggala and Netrapalli [2020, Page 4], our FTPL-D achieves an $O(T^{\frac{2}{3}}(V_T + 1)^{\frac{1}{3}})$ dynamic regret bound that matches the existing bound for convex functions [Besbes et al., 2015].

## 4.2 FTPL-D+

The dynamic regret in Theorem 1 requires the prior knowledge of $V_T$ to choose the optimal parameter $\gamma$. However, in practice, $V_T$ is usually unknown, posing a challenge in selecting the best $\gamma$. Studies on OCO also encounter the problem of searching for the optimal parameter for their algorithms [van Erven and Koolen, 2016, Zhang et al., 2018a, Wan et al., 2021, 2024]. The main idea in their solutions is activating multiple instances of their algorithms as experts, and tracking the best one with a meta-algorithm such as Hedge [Cesa-Bianchi and Lugosi, 2006], which assigns a weight to each expert and computes the weighted average of their advice.

Inspired by the above idea, we choose FTPL-D as the expert algorithm and propose our second algorithm called improved FTPL-D (FTPL-D+). Note that both the loss functions and the feasible set are non-convex, so we cannot combine different experts' predictions by the weighted average. According to the discussion of Cesa-Bianchi and Lugosi [2006, Chapter 4.1], we can tackle this issue by randomized sampling, where we sample one expert according to the weights, and then output the prediction of that expert.

The details of FTPL-D+ are shown in Algorithm 3, and we describe the main steps below. We first define a set $\mathcal{H} = \{\gamma_1, ..., \gamma_N\}$ of $N$ values for parameter $\gamma$. Then we activate a set of experts $\{E_i \mid \gamma_i \in \mathcal{H}\}$ by invoking Algorithm 2 as $E_i = \text{FTPL-D}(\mathcal{K}, \gamma_i, \mathcal{O}_{\alpha,\beta}, \eta)$. For each expert $E_i$, we denote its weight at round $t$ as $w_t^i$, which is initiated as $w_1^i = 1/N$. At each round $t$, we receive a prediction $\mathbf{x}_t^i$ from the expert $E_i$. To utilize these predictions $\{\mathbf{x}_t^i \mid i \in [N]\}$, we select $\mathbf{x}_t^i$ with the probability $w_t^i$ and submit it as the output $\mathbf{x}_t$, i.e., $P\left(\mathbf{x}_t = \mathbf{x}_t^i\right) = w_t^i$. After the loss function is revealed, we update the weights according to the following rule [Cesa-Bianchi and Lugosi, 2006]:

$$w_{t+1}^i = \frac{w_t^i e^{-\rho f_t(\mathbf{x}_t^i)}}{\sum_{j=1}^N w_t^j e^{-\rho f_t(\mathbf{x}_t^j)}}, \tag{9}$$

where $\rho > 0$ is the step size. We present the dynamic regret of Algorithm 3 in the following theorem.

**Theorem 2.** *Let* $\mathcal{H} = \left\{\gamma_i = 2^i \mid i = 1, \cdots N\right\}$ *where* $N = \lfloor \log_2 T \rfloor$, *and* $\rho = \frac{1}{dDL}\sqrt{\frac{8\ln N}{T}}$. *Under Assumptions 1 and 2, Algorithm 3 ensures*

$$\mathbb{E}\left[R_D^*\right] \leq O\left((1 + \alpha\sqrt{T} + \beta T)T^{\frac{2}{3}}(V_T + 1)^{\frac{1}{3}}\right).$$

***Remark.*** Theorem 2 shows that our FTPL-D+ achieves the same order of dynamic regret bound as Algorithm 2 without the prior knowledge of $V_T$.

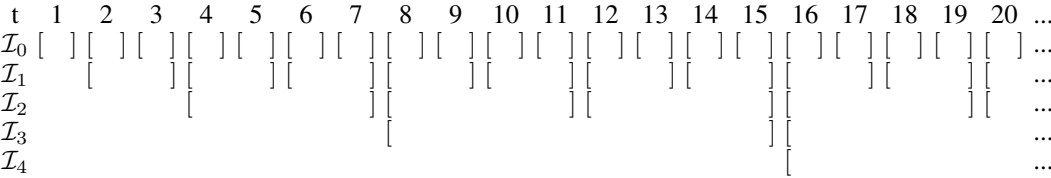

Figure 2: Graphic illustration of Geometric Covering intervals [Daniely et al., 2015] from Figure 1 of Zhang et al. [2019]. In the figure, each interval is denoted by [ ].

---

**Algorithm 4** FTPL-A

---

1: **for** $t = 1$ **to** $T$ **do**
2:     **for** $I \in \mathcal{C}_t$ **do**
3:         Create an expert $E_I$ which runs FTPL with $\eta = 1/\sqrt{d|I|}$
4:         For the expert $E_I$, set $R_{t-1,I} = C_{t-1,I} = 0$
5:         Add expert $E_I^S$ to the set of active experts $\mathcal{A}_t$
6:     **end for**
7:     Remove all experts who end at the round $t$ from $\mathcal{A}_t$
8:     Receive the predictions $\mathbf{x}_{t,I}$ of each expert $E_I \in \mathcal{A}_t$ and calculate its weight $w_{t,I}$ by (10)
9:     Draw $\mathbf{x}_t$ according to $P\left(\mathbf{x}_t = \mathbf{x}_{t,I}\right) = w_{t,I}$
10:    Output $\mathbf{x}_t$ and observe loss function $f_t(\cdot)$
11:    Compute the instantaneous expected loss defined in (11)
12:    For each expert $E_I \in \mathcal{A}_t$, update $R_{t,I}$ and $C_{t,I}$ by (12)
13:    Send loss function $f_t(\cdot)$ to each expert $E_I \in \mathcal{A}_t$
14: **end for**

---

# 5 Online Non-convex Learning with Adaptive Regret

In this section, we further consider strongly adaptive regret defined in (4) as the performance metric in dynamic environments. We develop our third algorithm, named follow the perturbed leader with adaptive regret (FTPL-A), and establish its adaptive and dynamic regret bounds.

Previous research on adaptive regret [Hazan and Seshadri, 2007, Daniely et al., 2015, Wang et al., 2024] has adopted a two-layered framework to design adaptive online algorithms. The basic idea is to dynamically construct a set of intervals and run an expert algorithm to minimize the static regret within each interval. These experts are then combined by a meta-algorithm. Building upon this idea, our FTPL-A includes three components: an expert algorithm, a set of intervals, and a meta-algorithm. In the following, we illustrate them separately.

First, we use FTPL as the expert algorithm, which enjoys $O(\sqrt{|I|})$ static regret for a given interval $I$ [Suggala and Netrapalli, 2020]. Then, following the work of Daniely et al. [2015], we build geometric covering (GC) intervals shown in Fig. 2:

$$\mathcal{I} = \bigcup_{k \in \mathbb{N}} \mathcal{I}_k, \quad \mathcal{I}_k = \left\{[i \cdot 2^k, (i+1) \cdot 2^k - 1] : i \in \mathbb{N}\right\}.$$

That is, each $\mathcal{I}_k$ is a partition of $\mathbb{N}\backslash\{1, \ldots, 2^k - 1\}$ into consecutive intervals of length $2^k$. For each interval $I \in \mathcal{I}$, we activate an instance of FTPL as the expert $E_I$ to minimize the regret over $I$.

Next, we choose AdaNormalHedge [Luo and Schapire, 2015] to track the best expert. Similar to the Hedge forecaster we used in FTPL-D+, we maintain a weight for each expert, but in a specific form of a potential function:

$$\Phi(R, C) = \exp\left(\frac{[R]_+^2}{3C}\right),$$

where $[x]_+ = \max(0, x)$ and $\Phi(0, 0)$ is defined to be 1, and the weight function is determined by

$$w(R, C) = \frac{1}{2}\left(\Phi(R+1, C+1) - \Phi(R+1, C-1)\right).$$

Due to the non-convexity of the loss functions, as explained in Section 4.2, we use randomized sampling to generate the output.

The procedure of FTPL-A is summarized in Algorithm 4. We describe the main steps below. For brevity, we denote the set of intervals starting from round $t$ as $\mathcal{C}_t$, and the set of active experts at round $t$ as $\mathcal{A}_t$. In Step 3, for each interval $I \in C_t$, we run an instance of FTPL as $E_I$ with the optimal parameters. In Step 4, the two variables $R_{t-1,I}$ and $C_{t-1,I}$ are initialized as 0 for $E_I$, where $R_{t-1,I}$ denotes the expected regret of $E_I$ up to round $t-1$, and $C_{t-1,I}$ denotes the expected absolute regret. In Steps 5 and 7, the active expert set $A_t$ is maintained by adding new experts and removing expired ones. In Step 8, the expert's weight is calculated by [Luo and Schapire, 2015]:

$$w_{t,I} = \frac{w(R_{t-1,I}, C_{t-1,I})}{\sum_{E_I \in \mathcal{A}_t} w(R_{t-1,I}, C_{t-1,I})}. \tag{10}$$

Similarly, we sample one expert (prediction) according to the weights in Step 9, and submit it as the output in Step 10. In Step 11, the instantaneous expected loss is computed as the weighted average of different experts' losses:

$$\tilde{f}_t(\mathbf{x}_t) = \sum_{E_I \in \mathcal{A}_t} w_{t,I} f_t(\mathbf{x}_{t,I}), \tag{11}$$

where $\mathbf{x}_{t,I}$ denotes the prediction of expert $E_I$ at round $t$. In Step 12, we update $R_{t,I}$ and $C_{t,I}$ by

$$R_{t,I} = R_{t-1,I} + \frac{\tilde{f}_t(\mathbf{x}_t) - f_t(\mathbf{x}_{t,I})}{dDL}, \quad C_{t,I} = C_{t-1,I} + \frac{|\tilde{f}_t(\mathbf{x}_t) - f_t(\mathbf{x}_{t,I})|}{dDL}. \tag{12}$$

The following theorem demonstrates the strongly adaptive regret bound of Algorithm 4 for general non-convex loss functions.

**Theorem 3.** *Under Assumptions 1 and 2, Algorithm 4 ensures*

$$\mathbb{E}\left[R_A(T, \tau)\right] \leq \left(8c(\alpha, \beta, \tau) + 8\sqrt{3dDLg(T)}\right)\sqrt{\tau} = O\left(\sqrt{\tau \log T} + \alpha\tau + \beta\tau^{\frac{3}{2}}\right),$$

*where $g(T) \leq 1 + \ln T + \ln(1 + \log_2 T) + \ln \frac{5 + 3\ln(1+T)}{2}$ and $c(\alpha, \beta, \tau)$ is given in (8).*

***Remark.*** Theorem 3 demonstrates that when $\alpha = O(1/\sqrt{\tau})$ and $\beta = O(1/\tau)$, FTPL-A is strongly adaptive with the same order of $O(\sqrt{\tau \log T})$ regret as that for convex functions [Jun et al., 2017].

Furthermore, we notice that Zhang et al. [2018b] investigated the relationship between adaptive regret and dynamic regret. They showed that dynamic regret can be bounded by the strongly adaptive regret and the functional variation. According to their findings, we have the following theorem that presents the dynamic regret bound of Algorithm 4.

**Theorem 4.** *Under Assumptions 1 and 2, Algorithm 4 ensures*

$$\mathbb{E}\left[R_D^*\right] \leq \tilde{O}\left((1 + \alpha\sqrt{T} + \beta T)T^{\frac{2}{3}}(V_T + 1)^{\frac{1}{3}}\right).$$

***Remark.*** Theorem 4 shows that FTPL-A achieves nearly the same dynamic regret as FTPL-D+, indicating its ability to minimize adaptive and dynamic regret simultaneously. However, compared to FTPL-A, FTPL-D+ does not require the construction of GC intervals and uses a simpler meta-algorithm, making it easier to implement.

## 6 Application to Online Constrained Meta-Learning

Online non-convex learning offers a wide range of applications [Neel et al., 2020, Vietri et al., 2020, Ghai et al., 2021, Castiglioni et al., 2022]. By applying our methods, we can extend these applications to dynamic environments. In this section, we discuss the application to online constrained meta-learning [Xu and Zhu, 2023] and conduct experiments to support our theoretical results.

**Online constrained meta-learning.** Meta-learning, also known as learning-to-learn, focuses on acquiring a prior meta-parameter that enables fast adaptation to new tasks. Recently, Xu and Zhu [2023] proposed the setting of online constrained meta-learning, which aims to learn the meta-parameter from a sequence of constrained learning tasks $\{\mathcal{T}_1, \ldots, \mathcal{T}_T\}$. Each task $\mathcal{T}_t$ is characterized by its data distributions $\mathcal{D}_t$ and constraint limits $c_t$. At each round $t$, a training dataset $\mathcal{D}_t^{tr}$, which contains data sampled i.i.d. from $\mathcal{D}_t$, is available to the learner. Then the learner adapts the meta-parameter $\phi_t$ to the task-specific parameter $\theta_t$ by a within-task algorithm $\mathcal{A}lg$ with $\mathcal{D}_t^{tr}$ and $c_t$. After

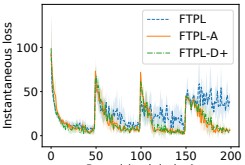 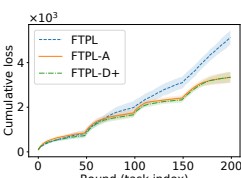 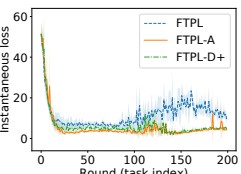 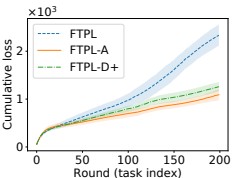

(a) Env1: instantaneous loss (b) Env1: cumulative loss (c) Env2: instantaneous loss (d) Env2: cumulative loss

Figure 3: Results of meta-imitation learning in two types of dynamic environments. The first two figures are for abruptly changing environments (Env1), while the latter ones are for gradually evolving environments (Env2).

the learner deploys $\theta_t$, it obtains a validation dataset $\mathcal{D}_t^{val}$ by sampling data from $\mathcal{D}_t$ and defines the meta-objective function at current round as

$$\mathcal{L}_t^{val}(\phi) = \mathcal{L}(Alg(\phi, \mathcal{D}_t^{tr}, c_t), \mathcal{D}_t^{val}),$$

where $\mathcal{L}(\theta, \mathcal{D}_t^{val})$ denotes the loss of $\theta$ on $\mathcal{D}_t^{val}$. Xu and Zhu [2023] updated the meta-parameter by using FTPL to the non-convex meta-objective function on all revealed tasks. With this approach, they effectively controlled the static regret of the meta-objective function between $\phi_t$ and the fixed optimal meta-parameter $\phi^*$. However, when the distribution of tasks is shifting, the fixed $\phi^*$ may not be adapted to each task well. In such a scenario, it is better to use dynamic regret or adaptive regret as the performance metric, since they measures the learner's performance against the changing optimal meta-parameter. By applying Algorithms 3 and 4, we can minimize the two metrics efficiently. To evaluate our methods, we conduct experiments on a sequence of tasks in imitation learning and another in few-shot learning, respectively.

## 6.1 Experiments on meta-imitation learning

**Setup.** We use the demonstration data given by Huang et al. [2019] and set the total number of tasks $T = 200$. At each round $t \in [T]$, a human expert writes a different letter in a free space without obstacle. The learner can observe a demonstration of the letter and is asked to write the same letter in a cluttered environment. We provide the details of problem formulation and implement setting in Appendix C.1. By controlling the letter to be imitated for each task, we simulate two types of dynamic environments: (i) *abruptly changing* environments; (ii) *gradually evolving* environments. In (i), we split the time horizon evenly into 4 stages, and set the imitation target as capital letters "M", "E", "T" and "A" for the 4 stages. In this way, the optimal meta-parameter $\phi^*$ drifts notably every 50 round. In (ii), we choose the capital letter "A" as the imitation target. Additionally, we rotate the letter "A" at a small random angle $\delta_t \in (0, 0.05]$ in each round. This ensures the optimal meta-parameter $\phi^*$ undergoes slow and smooth shifts. We compare our FTPL-D+ and FTPL-A with FTPL [Suggala and Netrapalli, 2020] in the above two scenarios.

**Results.** We repeat the experiments five times with different random seeds and plot the loss (mean and standard deviation) in Fig. 3. More results are provided in Appendix C.1. For abruptly changing environments, Fig. 3(a) shows that, in comparison to FTPL, our methods adapt more quickly to the new task distribution after the demonstration shifts, which occurs at $T = 50$, 100, and 150. Fig. 3(b) demonstrates that our methods perform significantly better than FTPL in terms of cumulative loss. For gradually evolving environments, it can be observed from Fig. 3(c) that as the angle of rotation increases, both our methods maintain low loss, while FTPL exhibits a significant increase in instantaneous loss after $T = 100$. Fig. 3(d) also shows a notable advantage in cumulative loss. Moreover, our FTPL-D+ and FTPL-A achieve comparable performance in the two types of dynamic environments.

## 6.2 Experiments on few-shot image classification with robustness

**Setup.** We conduct the experiments on CUB-200-2011 (referred to as CUB) dataset [Wah et al., 2011], which includes 200 fine-grained categories of birds. There are $T = 150$ of robust 5-way 5-shot image classification tasks, each task containing images from 5 classes with only 5 training samples per class. At each round $t \in [T]$, the model is updated using a few data samples and is required to have high accuracy on both clean and perturbed test data. We provide the details of problem

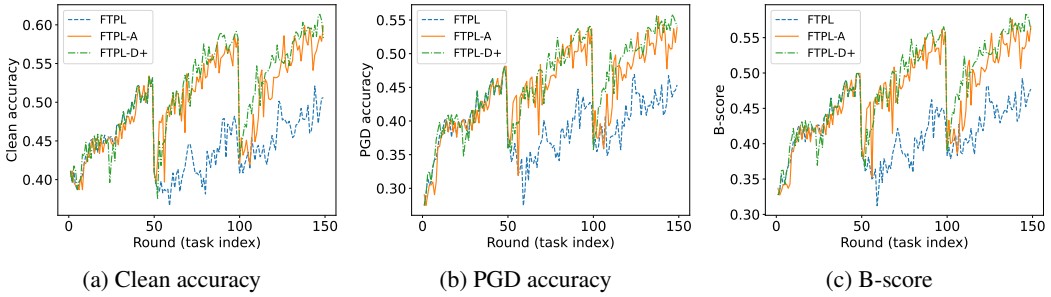

|                | (a) Clean accuracy | (b) PGD accuracy | (c) B-score |
|----------------|--------------------|------------------|-------------|

Figure 4: Results of robust few-shot learning on the CUB dataset (5-way 5-shot) in dynamic environments.

formulation and implement setting in Appendix C.2. We resize the input images to 84 × 84, and apply the same data augmentation as in Ye et al. [2021] and Xu and Zhu [2023]. A four-layer convolutional neural network (Conv-4) is employed as the backbone, comprising four blocks. Each block consists of a convolutional layer with 64 kernels of size 3 × 3, stride 1, and zero padding, followed by a batch normalization layer, a ReLU activation function, and lastly a 2 × 2 max-pooling layer. Following the convolutional layers, the network uses a fully connected linear layer with 5 neurons as a classifier to output the prediction for the input image. To simulate real-world dynamic environments, we select three groups of bird categories from the CUB dataset based on their habitats: water birds, forest birds, and grassland birds. We then split the time horizon $T$ evenly into 3 stages, within each stage the tasks are sampled from one group of bird categories. This simulates *abruptly changing* environments where the optimal meta-parameter $\phi^*$ drifts every 50 round. We compare our FTPL-D+ and FTPL-A with FTPL [Suggala and Netrapalli, 2020].

**Metrics and results.** The performance of the task-specific model is evaluated by: (i) clean accuracy; (ii) PGD accuracy; (iii) B-score. The clean accuracy is the accuracy on the clean test dataset, and PGD accuracy is the accuracy on the corrupted test dataset, which is obtained by adding perturbation on the clean test dataset by the Projected Gradient Descent (PGD) method [Kurakin et al., 2018]. Balance Score (B-score) [Ye et al., 2021] measures both clean accuracy and PGD accuracy. It is defined as B-score $= 2 \times (\text{CA} \times \text{PA})/(\text{CA} + \text{PA})$, where CA and PA denote clean accuracy and PGD accuracy respectively. We report the three metrics against the number of tasks in Fig. 3. As evidenced by the results, FTPL-D+ and FTPL-A attain comparable performance, and both significantly outperform FTPL in terms of all three metrics after the environmental changes at $T = 50$ and $100$. This suggests that our methods are effective in rapidly adapting to the new task distribution.

## 7  Conclusion and Future Work

This paper investigates online non-convex learning in dynamic environments, using dynamic regret and adaptive regret as performance metrics. For dynamic regret minimization, we propose FTPL-D with an $O(T^{\frac{2}{3}}(V_T + 1)^{\frac{1}{3}})$ regret bound. To eliminate the dependence on prior knowledge of $V_T$, we propose FTPL-D+, which runs multiple instances of FTPL-D and uses a meta-algorithm to track the best one. For adaptive regret minimization, we propose FTPL-A with a regret bound of $O(\sqrt{\tau \log T})$. Finally, we discuss the application to online constrained meta-learning, and the conducted experiments verify the effectiveness of our methods.

Currently, we bound the dynamic regret by the functional variation. A natural problem is whether we can derive regret bounds based on other regularities, such as the path length of comparators that is widely used in prior works for non-stationary online convex optimization [Zhang et al., 2018a, Zhao et al., 2020, 2024, Baby and Wang, 2021, Cutkosky, 2020]. This is left as a future work to explore.

## Acknowledgements

This work was partially supported by NSFC (62361146852, 62122037), and the Collaborative Innovation Center of Novel Software Technology and Industrialization.

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

# A Theoretical Analysis

## A.1 Proof of Theorem 1

Let $r = \frac{T}{\gamma}$, $q_i = (i-1)\gamma + 1$ for $i = 1, \ldots, r$, and $q_{r+1} = T + 1$. First, we decompose the dynamic regret as

$$
\begin{aligned}
R_D^* &= \sum_{t=1}^{T} f_t(\mathbf{x}_t) - \sum_{t=1}^{T} \min_{\mathbf{x} \in \mathcal{K}} f_t(\mathbf{x}) \\
&= \sum_{i=1}^{r} \left( \sum_{t=q_i}^{q_{i+1}-1} f_t(\mathbf{x}_t) - \sum_{t=q_i}^{q_{i+1}-1} \min_{\mathbf{x} \in \mathcal{K}} f_t(\mathbf{x}) \right) \\
&= \sum_{i=1}^{r} \left( \sum_{t=q_i}^{q_{i+1}-1} f_t(\mathbf{x}_t) - \min_{\mathbf{x} \in \mathcal{K}} \sum_{t=q_i}^{q_{i+1}-1} f_t(\mathbf{x}) + \min_{\mathbf{x} \in \mathcal{K}} \sum_{t=q_i}^{q_{i+1}-1} f_t(\mathbf{x}) - \sum_{t=q_i}^{q_{i+1}-1} \min_{\mathbf{x} \in \mathcal{K}} f_t(\mathbf{x}) \right). \quad (13)
\end{aligned}
$$

Taking expectation with respect to $\mathbf{x}_t$ on both sides of (13), we have

$$
\begin{aligned}
&\mathbb{E}\left[ R_D^* \right] \\
&= \mathbb{E}\left[ \sum_{i=1}^{r} \left( \sum_{t=q_i}^{q_{i+1}-1} f_t(\mathbf{x}_t) - \min_{\mathbf{x} \in \mathcal{K}} \sum_{t=q_i}^{q_{i+1}-1} f_t(\mathbf{x}) + \min_{\mathbf{x} \in \mathcal{K}} \sum_{t=q_i}^{q_{i+1}-1} f_t(\mathbf{x}) - \sum_{t=q_i}^{q_{i+1}-1} \min_{\mathbf{x} \in \mathcal{K}} f_t(\mathbf{x}) \right) \right] \\
&= \sum_{i=1}^{r} \left( \underbrace{\mathbb{E}\left[ \sum_{t=q_i}^{q_{i+1}-1} f_t(\mathbf{x}_t) - \min_{\mathbf{x} \in \mathcal{K}} \sum_{t=q_i}^{q_{i+1}-1} f_t(\mathbf{x}) \right]}_{a_i} + \underbrace{\min_{\mathbf{x} \in \mathcal{K}} \sum_{t=q_i}^{q_{i+1}-1} f_t(\mathbf{x}) - \sum_{t=q_i}^{q_{i+1}-1} \min_{\mathbf{x} \in \mathcal{K}} f_t(\mathbf{x})}_{b_i} \right). \quad (14)
\end{aligned}
$$

To bound $a_i$, we introduce the following lemma, which presents the static regret of Algorithm 1.

**Lemma 1.** *(Theorem 1 in Suggala and Netrapalli [2020]) Suppose Algorithm 1 is using an $(\alpha, \beta)$-approximate optimization oracle. Under Assumptions 1 and 2, for any fixed $\eta$, Algorithm 1 ensures*

$$
\mathbb{E}\left[ \sum_{t=1}^{T} f_t(\mathbf{x}_t) - \min_{\mathbf{x} \in \mathcal{K}} \sum_{t=1}^{T} f_t(\mathbf{x}) \right] \leq 125 T \eta d^2 D L^2 + \frac{d(21\beta T + D)}{20\eta} + \frac{21\alpha T}{20} + 2\beta dLT.
$$

By applying Lemma 1 to each interval $i \in [r]$, we get

$$
\mathbb{E}\left[ \sum_{t=q_i}^{q_{i+1}-1} f_t(\mathbf{x}_t) - \min_{\mathbf{x} \in \mathcal{K}} \sum_{t=q_i}^{q_{i+1}-1} f_t(\mathbf{x}) \right] \leq 125 \gamma \eta d^2 D L^2 + \frac{d(21\beta\gamma + D)}{20\eta} + \frac{21\alpha\gamma}{20} + 2\beta dL\gamma.
$$

Let $\eta = \frac{1}{\sqrt{d\gamma}}$, then we have

$$
a_i = \mathbb{E}\left[ \sum_{t=q_i}^{q_{i+1}-1} f_t(\mathbf{x}_t) - \min_{\mathbf{x} \in \mathcal{K}} \sum_{t=q_i}^{q_{i+1}-1} f_t(\mathbf{x}) \right] \leq c(\alpha, \beta, \gamma)\sqrt{\gamma}, \quad (15)
$$

where $c(\alpha, \beta, \gamma) = 125 D L^2 d^{\frac{3}{2}} + \frac{(21\beta\gamma + D)d^{\frac{3}{2}}}{20} + \frac{21\alpha\sqrt{\gamma}}{20} + 2dL\beta\gamma$.

To bound $b_i$, we follow the proof of Theorem 3 in Zhang et al. [2018b]:

$$
\begin{aligned}
b_i &= \min_{\mathbf{x} \in \mathcal{K}} \sum_{t=q_i}^{q_{i+1}-1} f_t(\mathbf{x}) - \sum_{t=1}^{T} \min_{\mathbf{x} \in \mathcal{K}} f_t(\mathbf{x}_t) \\
&\leq \sum_{t=q_i}^{q_{i+1}-1} f_t(\mathbf{x}_{q_i}^*) - \sum_{t=q_i}^{q_{i+1}-1} f_t(\mathbf{x}_t^*) \\
&\leq |q_{i+1} - q_i| \max_{t \in [q_i, q_{i+1}-1]} \left( f_t(\mathbf{x}_{q_i}^*) - f_t(\mathbf{x}_t^*) \right), \quad (16)
\end{aligned}
$$

where $\mathbf{x}_t^* \in \arg\min_{\mathbf{x} \in \mathcal{K}} f_t(\mathbf{x})$.

Define the local functional variation of the $i$-th interval as

$$V_T(i) = \sum_{t=q_i}^{q_{i+1}-1} \max_{\mathbf{x} \in \mathcal{K}} |f_t(\mathbf{x}) - f_{t-1}(\mathbf{x})|,$$

and it is obvious that $\sum_{i=1}^r V_T(i) \leq V_T$.

For any $t \in [q_i, q_{i+1} - 1]$, we have

$$
\begin{aligned}
f_t(\mathbf{x}_{q_i}^*) - f_t(\mathbf{x}_t^*) =& f_t(\mathbf{x}_{q_i}^*) - f_{q_i}(\mathbf{x}_{q_i}^*) + f_{q_i}(\mathbf{x}_{q_i}^*) - f_t(\mathbf{x}_t^*) \\
\leq& f_t(\mathbf{x}_{q_i}^*) - f_{q_i}(\mathbf{x}_{q_i}^*) + f_{q_i}(\mathbf{x}_t^*) - f_t(\mathbf{x}_t^*) \\
\leq& 2 V_T(i).
\end{aligned}
\tag{17}
$$

Combining (16) and (17), we obtain

$$b_i \leq 2 \left| q_{i+1} - q_i \right| V_T(i) \leq 2\gamma V_T(i). \tag{18}$$

Substituting (15) and (18) into (14), we have

$$
\begin{aligned}
\mathbb{E}\left[ R_D^* \right] &\leq \sum_{i=1}^r \left[ c(\alpha, \beta, \gamma)\sqrt{\gamma} + 2\gamma V_T(i) \right] \\
&\leq c(\alpha, \beta, \gamma) \left\lceil \frac{T}{\gamma} \right\rceil \sqrt{\gamma} + \sum_{i=1}^r 2\gamma V_T(i) \\
&\leq \frac{c(\alpha, \beta, \gamma) 2T}{\sqrt{\gamma}} + 2\gamma V_T
\end{aligned}
\tag{19}
$$

where the last inequality uses $\sum_{i=1}^r V_T(i) \leq V_T$ and $r = \left\lceil \frac{T}{\gamma} \right\rceil \leq \frac{T}{\gamma} + 1 \leq \frac{2T}{\gamma}$.

We proceed to prove the second part of Theorem 1. For brevity, let $\hat{c} = c(\alpha, \beta, T)$. By the definition in (8), we have $c(\alpha, \beta, \gamma) \leq \hat{c}$ for any $\gamma \leq T$. If $V_T \geq \frac{1}{\sqrt{T}}$, Algorithm 2 with $\gamma = \left\lfloor \left( \frac{T}{V_T} \right)^{\frac{2}{3}} \right\rfloor$ achieves

$$\mathbb{E}\left[ R_D^* \right] \leq \left( 2\sqrt{2}\hat{c} + 2 \right) T^{\frac{2}{3}} V_T^{\frac{1}{3}}.$$

If $V_T \leq \frac{1}{\sqrt{T}}$, Algorithm 2 with $\gamma = T$ ensures

$$\mathbb{E}\left[ R_D^* \right] \leq \left( 2\hat{c} + 2 \right) \sqrt{T}.$$

Combining the results above, with the prior knowledge of $V_T$, we have

$$\mathbb{E}\left[ R_D^* \right] \leq \max \left\{ \left( 2\sqrt{2}\hat{c} + 2 \right) T^{\frac{2}{3}} V_T^{\frac{1}{3}}, \left( 2\hat{c} + 2 \right) \sqrt{T} \right\} = O\left( (1 + \alpha\sqrt{T} + \beta T) T^{\frac{2}{3}} (V_T + 1)^{\frac{1}{3}} \right).$$

### A.2 Proof of Theorem 2

The dynamic regret of the Algorithm 3 can be devided into two parts: expert regret and meta-regret, as shown below

$$
\begin{aligned}
\mathbb{E}\left[ R_D^* \right] =& \mathbb{E}\left[ \sum_{t=1}^T f_t(\mathbf{x}_t) - \sum_{t=1}^T \min_{\mathbf{x} \in \mathcal{K}} f_t(\mathbf{x}) \right] \\
=& \underbrace{\mathbb{E}\left[ \sum_{t=1}^T f_t(\mathbf{x}_t) - \sum_{t=1}^T f_t(\mathbf{x}_t^i) \right]}_{\text{meta-regret}} + \underbrace{\mathbb{E}\left[ \sum_{t=1}^T f_t(\mathbf{x}_t^i) - \sum_{t=1}^T \min_{\mathbf{x} \in \mathcal{K}} f_t(\mathbf{x}) \right]}_{\text{expert regret}},
\end{aligned}
\tag{20}
$$

where $\mathbf{x}_t^i$ denotes the prediction of the $i$-th expert.

To bound the meta-regret, we follow the proof of Theorem 2.2 in Cesa-Bianchi and Lugosi [2006] and introduce a novel quantity $\ln \frac{W_T}{W_0}$. We define

$$
\begin{cases}
W_t = \sum_{i=1}^{N} e^{-\rho L_t^i}, & t \geq 1 \\
W_0 = N,
\end{cases}
$$

where $L_t^i = \sum_{\tau=1}^{t} f_\tau(\mathbf{x}_\tau^i)$ denotes the cumulative loss of the $i$-th expert and $N$ denotes the number of experts. Besides, the weight of expert at round $t$ can be writen as

$$
w_t^i = \frac{e^{-\rho L_{t-1}^i}}{\sum_{j=1}^{N} e^{-\rho L_{t-1}^j}}. \tag{21}
$$

First, we give a lower bound of $\ln \frac{W_T}{W_0}$:

$$
\begin{aligned}
\ln \frac{W_T}{W_0} &= \ln \left( \sum_{i=1}^{N} e^{-\rho L_T^i} \right) - \ln N \\
&\geq \ln \left( \max_{i=1,\ldots,N} e^{-\rho L_T^i} \right) - \ln N \\
&= -\rho \min_{i=1,\ldots,N} \sum_{t=1}^{T} f_t(\mathbf{x}_t^i) - \ln N. \tag{22}
\end{aligned}
$$

To upper bound $\ln \frac{W_T}{W_0}$, we introduce the following lemma of Hoeffding's inequality.

**Lemma 2.** *(Lemma A.1 in Cesa-Bianchi and Lugosi [2006]) Let X be a random variable with $a \leq X \leq b$. Then for any $s \in \mathbb{R}$,*

$$
\ln \mathbb{E} \left[ e^{sX} \right] \leq s \mathbb{E} X + \frac{s^2 (b-a)^2}{8}. \tag{23}
$$

Note that at each round $t$, we sample an expert (prediction) by $P\left(\mathbf{x}_t = \mathbf{x}_t^i\right) = w_t^i$, and incur the loss of that expert. Hence, the loss incurred by the meta-algorithm is a random variable $f_t(\mathbf{x}_t)$ with probability $P\left[f_t(\mathbf{x}_t) = f_t(\mathbf{x}_t^i)\right] = w_t^i$, and the expected loss is

$$
\mathbb{E}\left[f_t(\mathbf{x}_t)\right] = \sum_{i=1}^{N} w_t^i f_t(\mathbf{x}_t^i). \tag{24}
$$

Under Assumptions 1 and 2, the difference between the maximum and minimum of the loss function at each round can be bounded as

$$
\sup_{\mathbf{x},\mathbf{y}\in\mathcal{K}} |f_t(\mathbf{x}) - f_t(\mathbf{y})| \leq dDL. \tag{25}
$$

Observe that for each $t = 1, \ldots, T$,

$$
\ln \frac{W_t}{W_{t-1}} = \ln \frac{\sum_{i=1}^{N} e^{-\rho L_{t-1}^i} e^{-\rho f_t(\mathbf{x}_t^i)}}{\sum_{j=1}^{N} e^{-\rho L_{t-1}^j}} \stackrel{(21)}{=} \ln \sum_{i=1}^{N} w_t^i e^{-\rho f_t(\mathbf{x}_t^i)}.
$$

Now using Lemma 2, we have

$$
\begin{aligned}
\ln \frac{W_t}{W_{t-1}} &= \ln \sum_{i=1}^{N} w_t^i e^{-\rho f_t(\mathbf{x}_t^i)} \\
&\stackrel{(23,25)}{\leq} -\rho \sum_{i=1}^{N} w_t^i f_t(\mathbf{x}_t^i) + \frac{\rho^2 (dDL)^2}{8} \\
&\stackrel{(24)}{=} -\rho \mathbb{E}[f_t(\mathbf{x}_t)] + \frac{\rho^2 (dDL)^2}{8}. \tag{26}
\end{aligned}
$$

Summing over $t = 1, \ldots, T$, we have

$$\ln \frac{W_t}{W_0} \leq -\rho \mathbb{E}\left[\sum_{t=1}^{T} f_t(\mathbf{x}_t)\right] + \frac{T\rho^2 (dDL)^2}{8}. \tag{27}$$

By combining (22) and (27), we get the meta-regret-bound as follows:

$$\mathbb{E}\left[\sum_{t=1}^{T} f_t(\mathbf{x}_t) - \min_{i=1\ldots N} \sum_{t=1}^{T} f_t(\mathbf{x}_t^i)\right] \leq \frac{\ln N}{\rho} + \frac{\rho T (dDL)^2}{8} \leq dDL\sqrt{\frac{\ln NT}{2}}, \tag{28}$$

where $\rho = \frac{1}{dDL}\sqrt{\frac{8\ln N}{T}}$.

We proceed to present the dynamic regret of the best expert. Let $\mathcal{H} = \left\{\gamma_i = 2^i \mid i = 1, \cdots N\right\}$ where $N = \lfloor \log_2 T \rfloor$. Assuming the optimal restarting parameter in Theorem 1 is $\gamma^*$, then there exists some $k$ such that $\gamma_k \leq \gamma^* \leq \gamma_{k+1}$. Moreover, we have

$$\frac{\gamma^*}{2} \leq \gamma_k \leq \gamma^* \leq \gamma_{k+1} \leq 2\gamma^*. \tag{29}$$

According to Theorem 1, for any expert $i$, we have

$$\mathbb{E}\left[R_D^i\right] = \mathbb{E}\left[\sum_{t=1}^{T} f_t(\mathbf{x}_t^i) - \sum_{t=1}^{T} \min_{\mathbf{x} \in \mathcal{K}} f_t(\mathbf{x})\right] \leq \frac{2\hat{c}T}{\sqrt{\gamma_i}} + 2\gamma_i V_T, \tag{30}$$

where $\hat{c} = c(\alpha, \beta, T)$ is given in (8).

Then, for expert $k$ and $k + 1$,

$$\mathbb{E}\left[R_D^k\right] \overset{(30)}{\leq} \frac{2\hat{c}T}{\sqrt{\gamma_k}} + 2\gamma_k V_T \leq \frac{2\hat{c}T}{\sqrt{\frac{1}{2}\gamma^*}} + 2\gamma^* V_T \overset{(29)}{\leq} \sqrt{2}\left(\frac{2\hat{c}T}{\sqrt{\gamma^*}} + 2\gamma^* V_T\right), \tag{31}$$

and

$$\mathbb{E}\left[R_D^{k+1}\right] \overset{(30)}{\leq} \frac{2\hat{c}T}{\sqrt{\gamma_{k+1}}} + 2\gamma_{k+1} V_T \leq \frac{2\hat{c}T}{\sqrt{\gamma^*}} + 4\gamma^* V_T \overset{(29)}{\leq} 2\left(\frac{2\hat{c}T}{\sqrt{\gamma^*}} + 2\gamma^* V_T\right). \tag{32}$$

From Theorem 1, it can be inferred that

$$\mathbb{E}\left[\widehat{R}_D^*\right] \leq \frac{2\hat{c}T}{\sqrt{\gamma^*}} + 2\gamma^* V_T \leq \max\left\{(2\sqrt{2}\hat{c} + 2)T^{\frac{2}{3}} V_T^{\frac{1}{3}}, (2\hat{c} + 2)\sqrt{T}\right\}. \tag{33}$$

Hence, the dynamic regret of the best expert can be bounded as

$$\begin{aligned}
\min_{i=1\ldots N} \mathbb{E}\left[R_D^i\right] &\leq \min\left\{\mathbb{E}\left[R_D^k\right], \mathbb{E}\left[R_D^{k+1}\right]\right\} \\
&\overset{(31,32)}{\leq} \sqrt{2}\left(\frac{2\hat{c}T}{\sqrt{\gamma^*}} + 2\gamma^* V_T\right) \\
&\overset{(33)}{\leq} \max\left\{(4\hat{c} + 2\sqrt{2})T^{\frac{2}{3}} V_T^{\frac{1}{3}}, (2\sqrt{2}\hat{c} + 2\sqrt{2})\sqrt{T}\right\}.
\end{aligned} \tag{34}$$

Substituting (28) and (34) into (20), we get the dynamic regret of Algorithm 3:

$$\begin{aligned}
\mathbb{E}\left[R_D^*\right] &\leq \max\left\{(4\hat{c} + 2\sqrt{2})T^{\frac{2}{3}} V_T^{\frac{1}{3}}, (2\sqrt{2}\hat{c} + 2\sqrt{2})\sqrt{T}\right\} + dDL\sqrt{\frac{\ln NT}{2}} \\
&= O\left((1 + \alpha\sqrt{T} + \beta T)T^{\frac{2}{3}} (V_T + 1)^{\frac{1}{3}}\right).
\end{aligned}$$

### A.3 Proof of Theorem 3

The analysis consists of two parts. We first upper bound the strongly adaptive regret over any interval $J = [i, j] \in \mathcal{I}$. Then, we extend the regret bound to any interval $I = [s, s + \tau - 1] \subseteq [T]$.

For any interval $J = [i, j] \in \mathcal{I}$, we can decompose the regret as

$$
\mathbb{E}\left[\sum_{t=i}^{j} f_t(\mathbf{x}_t) - \min_{\mathbf{x} \in \mathcal{K}} \sum_{t=i}^{j} f_t(\mathbf{x})\right]
$$

$$
= \mathbb{E}\left[\sum_{t=i}^{j} f_t(\mathbf{x}_t) - \sum_{t=i}^{j} f_t(\mathbf{x}_{t,J}) + \sum_{t=i}^{j} f_t(\mathbf{x}_{t,J}) - \min_{\mathbf{x} \in \mathcal{K}} \sum_{t=i}^{j} f_t(\mathbf{x})\right]
$$

$$
= \underbrace{\mathbb{E}\left[\sum_{t=i}^{j} f_t(\mathbf{x}_t) - \sum_{t=i}^{j} f_t(\mathbf{x}_{t,J})\right]}_{\text{meta-regret}} + \underbrace{\mathbb{E}\left[\sum_{t=i}^{j} f_t(\mathbf{x}_{t,J}) - \min_{\mathbf{x} \in \mathcal{K}} \sum_{t=i}^{j} f_t(\mathbf{x})\right]}_{\text{expert regret}}, \tag{35}
$$

According to Lemma 1 and (15), we have the following expert regret

$$
\mathbb{E}\left[\sum_{t=i}^{j} f_t(\mathbf{x}_{t,J}) - \min_{\mathbf{x} \in \mathcal{K}} \sum_{t=i}^{j} f_t(\mathbf{x})\right] \leq c(\alpha, \beta, |J|)\sqrt{|J|}. \tag{36}
$$

Next, we analyze the meta-regret. Let $N_t$ denote the total number of experts that have been seen up to round $t$. Recall the definition of GC intervals:

$$
\mathcal{I} = \bigcup_{k \in \mathbb{N}} \mathcal{I}_k, \quad \mathcal{I}_k = \left\{[i \cdot 2^k, (i+1) \cdot 2^k - 1] : i \in \mathbb{N}\right\}.
$$

For every $k$ that satisfies $2^k < t$, we have that a single interval in $\mathcal{I}_k$ contains $t$. Since we activate an expert for each interval, the number of active experts at round $t$ is $1 + \lfloor \log_2 t \rfloor$. Taking each round into account, it is easy to verify that

$$
N_t \leq t(1 + \log_2 t). \tag{37}
$$

For interval $J$, we denote the regret at round $t$ as $r_{t,J}$. By (25), we scale $r_{t,J}$ to ensure

$$
r_{t,J} = \frac{1}{dDL}\left(\tilde{f}_t(\mathbf{x}_t) - f_t(\mathbf{x}_{t,J})\right) \in [-1, 1]. \tag{38}
$$

Then, according to Theorems 1 and 3 in Luo and Schapire [2015], we have the following lemma.

**Lemma 3.** *Under Assumptions 1 and 2, for any interval $J = [i, j] \in \mathcal{I}$, Algorithm 4 ensures*

$$
\sum_{t=i}^{j} r_{t,J} \leq \sqrt{3g(j)|J|}, \tag{39}
$$

*where*

$$
g(j) \leq 1 + \ln N_j + \ln \frac{5 + 3\ln(1+j)}{2} \overset{(37)}{\leq} 1 + \ln j + \ln(1 + \log_2 j) + \ln \frac{5 + 3\ln(1+j)}{2}.
$$

Combining (38) and (39), we obtain that

$$
\sum_{t=i}^{j} \tilde{f}_t(\mathbf{x}_t) - \sum_{t=i}^{j} f_t(\mathbf{x}_{t,J}) = dDL \sum_{t=i}^{j} r_{t,J} \leq dDL\sqrt{3g(j)|J|}, \tag{40}
$$

Note that at each round $t$, we sample $\mathbf{x}_t$ by $P(\mathbf{x}_t = \mathbf{x}_{t,J}) = w_{t,J}$. Thus, the expected loss incurred by the meta-algorithm is

$$
\mathbb{E}[f_t(\mathbf{x}_t)] = \sum_{E_J \in \mathcal{A}_t} w_{t,J} f_t(\mathbf{x}_{t,J}). \tag{41}
$$

Also note that $\tilde{f}_t(\mathbf{x}_t)$ is defined as the weighted average loss of all experts, which is of the same form as (41), i.e.

$$
\tilde{f}_t(\mathbf{x}_t) = \sum_{E_J \in \mathcal{A}_t} w_{t,J} f_t(\mathbf{x}_{t,J}). \tag{42}
$$

Replacing the weighted average loss in (40) by expected loss, we have

$$\mathbb{E}\left[\sum_{t=i}^{j} f_t(\mathbf{x}_t) - \sum_{t=i}^{j} f_t(\mathbf{x}_{t,J})\right] \le dDL\sqrt{3g(j)|J|}. \tag{43}$$

Combining the expert regret in (36) and the meta-regret in (43), we obtain that

$$\mathbb{E}\left[\sum_{t=i}^{j} f_t(\mathbf{x}_t) - \min_{\mathbf{x}\in\mathcal{K}}\sum_{t=i}^{j} f_t(\mathbf{x})\right] \le c(\alpha,\beta,|J|)\sqrt{|J|} + dDL\sqrt{3g(j)|J|}, \tag{44}$$

where $g(j) \le 1 + \ln j + \ln(1 + \log_2 j) + \ln\frac{5+3\ln(1+j)}{2}$.

In the following, we extend the regret bound over $J$ to any interval $I = [s, s+\tau-1] \subseteq [T]$. We first introduce a property of GC intervals as below.

**Lemma 4.** *(Lemma 1.2 in Daniely et al. [2015]) For any interval $I = [s, s+\tau-1] \subseteq [T]$, it can be partitioned into two sequences of disjoint and consecutive intervals, denoted by $I_{-p}, \ldots, I_0 \in \mathcal{D}$ and $I_1, \ldots, I_q \in \mathcal{D}$, such that*

$$|I_{-i}|/|I_{-i+1}| \le 1/2, \forall i \ge 1 \quad and \quad |I_i|/|I_{i-1}| \le 1/2, \forall i \ge 2$$

According to Lemma 4, we have that

$$\mathbb{E}\left[\sum_{t=r}^{r+\tau-1} f_t(\mathbf{x}_t) - \min_{\mathbf{x}\in\mathcal{K}}\sum_{t=r}^{r+\tau-1} f_t(\mathbf{x})\right] = \sum_{i=-p}^{q}\left(\mathbb{E}\left[\sum_{t\in I_i} f_t(\mathbf{x}_t) - \min_{\mathbf{x}\in\mathcal{K}}\sum_{t\in I_i} f_t(\mathbf{x})\right]\right)$$

$$\le \sum_{i=-p}^{q}\left(c(\alpha,\beta,|I_i|)\sqrt{|I_i|} + dDL\sqrt{3g(s+\tau-1)|I_i|}\right)$$

$$\le \sum_{i=-p}^{q}\left(c(\alpha,\beta,\tau)\sqrt{|I_i|} + dDL\sqrt{3g(s+\tau-1)|I_i|}\right)$$

$$\le 2\sum_{i=0}^{\infty}\left(c(\alpha,\beta,\tau)\sqrt{2^{-i}\tau} + dDL\sqrt{3g(s+\tau-1)2^{-i}\tau}\right)$$

$$\le \left(2c(\alpha,\beta,\tau) + 2dDL\sqrt{3g(s+\tau-1)}\right)\sqrt{\tau}\sum_{i=0}^{\infty}\sqrt{2^{-i}}$$

$$\le \left(8c(\alpha,\beta,\tau) + 8dDL\sqrt{3g(s+\tau-1)}\right)\sqrt{\tau}. \tag{45}$$

Hence, the strongly adaptive regret of Algorithm 4 is

$$\mathbb{E}[R_A(T,\tau)] = \max_{[s,s+\tau-1]\subseteq[T]}\left(\mathbb{E}\left[\sum_{t=r}^{r+\tau-1} f_t(\mathbf{x}_t) - \min_{\mathbf{x}\in\mathcal{K}}\sum_{t=r}^{r+\tau-1} f_t(\mathbf{x})\right]\right)$$

$$\le \left(8c(\alpha,\beta,\tau) + 8dDL\sqrt{3g(T)}\right)\sqrt{\tau} = O(\sqrt{\tau\log T} + \alpha\tau + \beta\tau^{\frac{3}{2}}), \tag{46}$$

where $g(T) \le 1 + \ln T + \ln(1 + \log_2 T) + \ln\frac{5+3\ln(1+T)}{2}$.

### A.4 Proof of Theorem 4

The analysis is similar to Corollary 5 in Zhang et al. [2018b], which exploits the following lemma to bound the dynamic regret by the strongly adaptive regret and the functional variation.

**Lemma 5.** *(Theorem 3 in Zhang et al. [2018b]) Let $\mathbf{u}_t^* \in \arg\min_{\mathbf{u}\in\mathcal{K}} f_t(\mathbf{u})$. For all integer $k \in [T]$, we have*

$$R_D(\mathbf{u}_1^*, \ldots, \mathbf{u}_T^*) \le \min_{\mathcal{I}_1,\ldots,\mathcal{I}_k}\sum_{i=1}^{k}\left(R_A(T,|\mathcal{I}_i|) + 2|\mathcal{I}_i|\cdot V_T(i)\right),$$

*where the minimization is taken over any sequence of intervals.*

For our randomized algorithm, we take the expectation over both sides

$$\mathbb{E}\left[R_D^*\right] = \mathbb{E}\left[R_D(\mathbf{u}_1^*, \ldots, \mathbf{u}_T^*)\right] \leq \min_{\mathcal{I}_1, \ldots, \mathcal{I}_k} \sum_{i=1}^{k} \left(\mathbb{E}\left[R_A(T, |\mathcal{I}_i|)\right] + 2|\mathcal{I}_i| \cdot V_T(i)\right).$$

Next, we restric to intervals of length $\tau$, and in this case $k = \frac{T}{\tau}$. Then we get

$$\mathbb{E}\left[R_D^*\right] \leq \min_{1 \leq \tau \leq T} \sum_{i=1}^{k} \left(\mathbb{E}\left[R_A(T, \tau)\right] + 2\tau V_T(i)\right)$$

$$= \min_{1 \leq \tau \leq T} \left(\frac{\mathbb{E}\left[R_A(T, \tau)\right] T}{\tau} + 2\tau \sum_{i=1}^{k} V_T(i)\right)$$

$$\leq \min_{1 \leq \tau \leq T} \left(\frac{\mathbb{E}\left[R_A(T, \tau)\right] T}{\tau} + 2\tau V_T\right). \tag{47}$$

Combining (46) and (47), we have

$$\mathbb{E}\left[R_D^*\right] \leq \min_{1 \leq \tau \leq T} \left(\frac{\left(8c(\alpha, \beta, \tau) + 8dDL\sqrt{3g(T)}\right) T}{\sqrt{\tau}} + 2\tau V_T\right).$$

By the fact that

$$g(T) \leq 1 + \ln T + \ln\left(1 + \log_2 T\right) + \ln \frac{5 + 3\ln\left(1 + T\right)}{2} \leq 3 + 3\log T,$$

we obtain

$$\mathbb{E}\left[R_D^*\right] \leq \min_{1 \leq \tau \leq T} \left(\frac{\left(8c(\alpha, \beta, \tau) + 24dDL\sqrt{1 + \log T}\right) T}{\sqrt{\tau}} + 2\tau V_T\right). \tag{48}$$

In the following, we consider two cases. If $V_T \geq \sqrt{\frac{\log T}{T}}$, we choose

$$\tau = \left(\frac{T\sqrt{\log T}}{V_T}\right)^{2/3} \leq T,$$

and have

$$\mathbb{E}\left[R_D^*\right] \leq \frac{\left(8\hat{c} + 24dDL\sqrt{1 + \log T}\right) T^{\frac{2}{3}} V_T^{\frac{1}{3}}}{\log^{\frac{1}{6}} T} + 2T^{\frac{2}{3}} V_T^{\frac{1}{3}} \log^{\frac{1}{3}} T$$

$$\leq \frac{\left(8\hat{c} + 24dDL\right) T^{\frac{2}{3}} V_T^{\frac{1}{3}}}{\log^{\frac{1}{6}} T} + \left(2 + 24dDL\right) T^{\frac{2}{3}} V_T^{\frac{1}{3}} \log^{\frac{1}{3}} T, \tag{49}$$

where $\hat{c} = c(\alpha, \beta, T)$.

Otherwise, we choose $\tau = T$, and have

$$\mathbb{E}\left[R_D^*\right] \leq \left(8\hat{c} + 24dDL\sqrt{1 + \log T}\right) \sqrt{T} + 2TV_T$$

$$\leq \left(8\hat{c} + 24dDL\sqrt{1 + \log T}\right) \sqrt{T} + 2T\sqrt{\frac{\log T}{T}}$$

$$\leq \left(8\hat{c} + (24dDL + 2)\sqrt{1 + \log T}\right) \sqrt{T}. \tag{50}$$

Combining (49) and (50), we have

$$\mathbb{E}\left[R_D^*\right] \leq \max \begin{cases} \frac{(8\hat{c} + 24dDL) T^{\frac{2}{3}} V_T^{\frac{1}{3}}}{\log^{\frac{1}{6}} T} + (2 + 24dDL) T^{\frac{2}{3}} V_T^{\frac{1}{3}} \log^{\frac{1}{3}} T \\ \left(8\hat{c} + (24dDL + 2)\sqrt{1 + \log T}\right) \sqrt{T} \end{cases} \tag{51}$$

$$= O\left(\max\left\{(1 + \alpha\sqrt{T} + \beta T)\sqrt{T \log T}, (1 + \alpha\sqrt{T} + \beta T) T^{\frac{2}{3}} V_T^{\frac{1}{3}} \log^{\frac{1}{3}} T\right\}\right). \tag{52}$$

# B  More Related Works

In OCO, the computational effective variants of FTPL are investigated in centralized [Hazan and Minasyan, 2020] and decentrailized [Wang et al., 2023] setting. In addition to the worst-case dynamic regret defined in (2), Zinkevich [2003] also introduce the general dynamic regret, which compares the learner against *any* sequence of comparators

$$R_D(\mathbf{u}_1, \ldots, \mathbf{u}_T) = \sum_{t=1}^{T} f_t(\mathbf{x}_t) - \sum_{t=1}^{T} f_t(\mathbf{u}_t), \tag{53}$$

where $\mathbf{u}_1, \ldots, \mathbf{u}_T \in \mathcal{K}$. In this pioneer work, Zinkevich [2003] proposed a regularity of comparator sequence called path-length

$$P_T(\mathbf{u}_1, \ldots, \mathbf{u}_T) = \sum_{t=2}^{T} \|\mathbf{u}_t - \mathbf{u}_{t-1}\|,$$

and showed that OGD attains an $O(\sqrt{T}(1 + P_T))$ general dynamic regret bound. Zhang et al. [2018a] strengthened the general dynamic regret bound to $O(\sqrt{T(1 + P_T)})$, which has been shown to be optimal. Improved results can be obtained when the online functions are smooth [Zhao et al., 2020, 2024], strongly convex [Baby and Wang, 2022] or exp-concave [Baby and Wang, 2021]. Moreover, the computational efficiency of general dynamic regret optimization is recently considered [Zhao et al., 2022, Wang et al., 2024].

Notice that by assigning $\mathbf{u}_t \in \arg\min_{\mathbf{x} \in \mathcal{K}} f_t(\mathbf{x})$ as the local minimizer, we get the worst-case dynamic regret in (2), and there exist numerous studies on the worst-case scenario. When the loss functions are strong convex and smooth [Mokhtari et al., 2016], or when they are smooth and the minimizers lie in the interior of $\mathcal{K}$ [Yang et al., 2016], OGD can attain $O(P_T^* + 1)$ dynamic regret bound, where $P_T^* := P_T(\mathbf{x}_1^*, \ldots, \mathbf{x}_T^*)$ and $\mathbf{x}_t^* \in \arg\min_{\mathbf{x} \in \mathcal{K}} f_t(\mathbf{x})$. Another regularity of comparator sequence is the squared path-length [Zhang et al., 2017]:

$$S_T^* := S_T(\mathbf{x}_1^*, \ldots, \mathbf{x}_T^*) = \sum_{t=2}^{T} \|\mathbf{x}_t^* - \mathbf{x}_{t-1}^*\|^2,$$

which could be much smaller than the path-length $P_T^*$ when local minimizers move slowly. Zhang et al. [2017] demonstrated that the dynamic regret bound can be reduced to $O(\min\{P_T^*, S_T^*\} + 1)$ for (semi-)strongly convex and smooth functions. Moreover, Besbes et al. [2015] demonstrated that if the value of $V_T$ is known, a restarted OGD can achieve $O(T^{\frac{2}{3}}(V_T + 1)^{\frac{1}{3}})$ and $O(\sqrt{T(V_T + 1)})$ dynamic regret for convex and strongly convex functions, respectively. Later, Baby and Wang [2019] improved the dynamic regret to $O(T^{\frac{1}{3}}(V_T + 1)^{\frac{2}{3}})$ for 1-dim square loss. More recently, to exploit both comparator sequence and function sequence, Zhao and Zhang [2021] refined the analysis for online multiple gradient descent and attained an $O(\min\{P_T^*, S_T^*, V_T\} + 1)$ dynamic regret bound under the same assumption as Zhang et al. [2017].

Besides dynamic regret, another metric in dynamic environments is adaptive regret. Adaptive regret is introduced by Hazan and Seshadhri [2007] to OCO, but in a weak form:

$$R_A(T) = \max_{[s,r] \subseteq [T]} \left\{ \sum_{t=s}^{r} f_t(\mathbf{x}_t) - \min_{\mathbf{x} \in \mathcal{K}} \sum_{t=s}^{r} f_t(\mathbf{x}) \right\}, \tag{54}$$

which is defined as the maximum static regret over any contiguous interval. To minimize (54), Hazan and Seshadhri [2007] proposed follow the leading history (FLH) with $O(\sqrt{T} \log^3 T)$ weakly adaptive regret for convex functions. However, this metric does not respect short intervals well since the $O(\sqrt{T} \log^3 T)$ bound is meaningless for those intervals of length $O(\sqrt{T})$. To eliminate the dominance of long intervals, Daniely et al. [2015] introduced strongly adaptive regret that takes the interval length $\tau$ as a parameter, as we indicate in (4). They also developed two-layer algorithms and attain $O(\sqrt{\tau} \log T)$ strongly adaptive regret for general convex functions. Later, Jun et al. [2017] proposed a novel meta-algorithm named sleeping coin betting (SCB) and improved the strongly adaptive regret bound to $O(\sqrt{\tau \log T})$.

Although both dynamic regret and adaptive regret are designed for dynamic environments, our understanding of their relationship remains limited. Zhang et al. [2018b] first demonstrated that

dynamic regret can be bounded by the strongly adaptive regret and the functional variation. Recent studies shows that it is possible to minimize dynamic regret and adaptive regret simultaneously [Zhang et al., 2020, Cutkosky, 2020, Zhang et al., 2022, Wang et al., 2024].

The metric of dynamic regret has also been investigated in online non-convex learning. Lesage-Landry et al. [2020] proposed online newton method (ONM), a second-order method, achieving an $O(P_T^* + 1)$ dynamic regret bound under the conditions that the starting point is located near the global optimal solution and the loss function is strongly convex around the optimal solution. Gao et al. [2018] examined a class of non-convex functions that satisfy weak pseudo-convex conditions and developed online algorithms that attain $O(\sqrt{T(V_T + 1)})$ dynamic regret. They also extended their result to the bandit feedback setting. Roy et al. [2019] studied functions with weak-quasi-convexity and obtained similar results as Gao et al. [2018]. Furthermore, Héliou et al. [2020] applied their methods to bandit setting using a kernel-based estimator. In a subsequent work, Héliou et al. [2021] proposed hierarchical dual averaging (HDA) and improved the dynamic regret bounds under the same setting.

# C   Omitted Details for Experiments

All experiments are executed on a computer with a 2.50 GHz Intel Xeon Platinum 8255C CPU and an RTX 2080Ti GPU. We follow the experiment settings of Xu and Zhu [2023] and implement our methods based on their code.

## C.1   Experiments on meta-imitation learning.

**Problem formulation.**   Imitation learning [Billard et al., 2004] has gained significant attention as a means of transferring human skills to robots. Huang et al. [2019] presents a novel formulation of imitation learning using kernelized movement primitives, which considers nonlinear hard constraints and obstacle avoidance. In their study, the states of the robot are modeled as a linear combination of basis functions, and the demonstrations provided by humans are modeled by a Gaussian mixture model (GMM), where the parameters follow a Gaussian distribution. The objective of their approach is to minimize the divergence between the distributions of the robot state model and the demonstrations, while ensuring that the hard constraints are satisfied. Building upon this idea, Xu and Zhu [2023] further model the states of the robot by a neural network instead of the linear combination of basis functions.

Following the work of Xu and Zhu [2023], in this experiment, we utilize a neural network to model the states of the robot and GMM to model the demonstrations. Specifically, the robot's state $\xi(w, t)$, which includes the joint position $q(w, t) \in R^{\mathcal{O}}$ and velocity $\dot{q}(w, t)$, is parameterized by $w$ and is modeled as

$$\xi(w, t) = \left[ \begin{array}{c} q(w, t) \\ \dot{q}(w, t) \end{array} \right],$$

where $q(w, t)$ is a neural network and takes $t$ as the input and the location $q(w, t)$ as the output. The dataset of demonstrations consists of $H$ trajectories, where each trajectory contains $N$ time-state pairs. These pairs, denoted as $\{\{t_{n,h}, \hat{\xi}_{n,h}\}_{n=1}^N\}_{h=1}^H$, are modeled using a GMM. Each demonstration state $\hat{\xi}_n$ associated with $t_n$ is described by a conditional probability distribution with mean $\hat{\mu}_n$ and covariance $\hat{\Sigma}_n$, i.e., $\xi_n \mid t_n \sim \mathcal{N}(\hat{\mu}_n, \hat{\Sigma}_n)$, where $\hat{\mu}_n$ and $\hat{\Sigma}_n$ can be computed by the GMM. As formulated in previous works [Huang et al., 2019, Xu and Zhu, 2023], the imitation learning task can be cast as the following constrained optimization problem:

$$\begin{aligned} \min_{w} \quad & \mathbb{E}_{t \in [t_0, t_N]} \left[ \frac{1}{2} (\xi(w, t) - \hat{\mu}_t)^\top \hat{\Sigma}_t^{-1} (\xi(w, t) - \hat{\mu}_t) \right] \\ \text{s.t.} \quad & g_i(\xi(w, t)) \leq c_i, \ \forall t \in [t_0, t_N], \ i = 1, \ldots, m, \end{aligned} \tag{55}$$

where $g_i$ is the i-th state constraint, $m$ is the total constraint number, $\hat{\mu}_t$ and $\hat{\Sigma}_t$ are the mean and variance of the demonstration state $\hat{\xi}(t)$, i.e., $\hat{\xi} \mid t \sim \mathcal{N}(\hat{\mu}_t, \hat{\Sigma}_t)$. Here, the training dataset $\{t_n, \hat{\mu}_n, \hat{\Sigma}_n\}_{n=1}^N$ is provided by the GMM. It is notable that the demonstrations are collected in a no-collision environment. Thus, it is possible that these demonstrations may not be able to avoid collisions effectively.

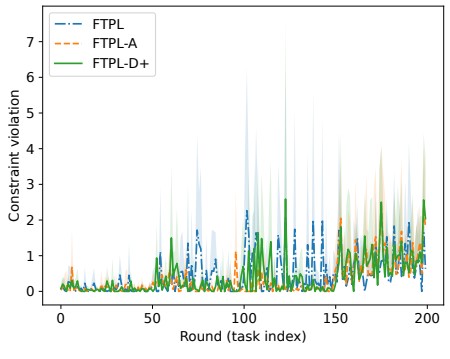

(a) Constrained violation metric

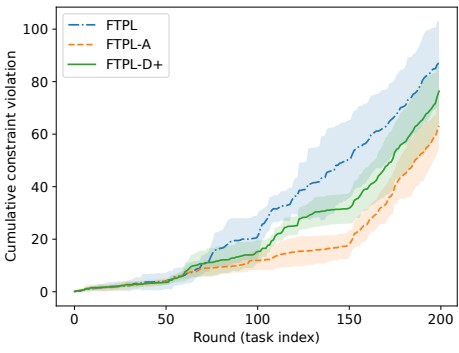

(b) Cumulative constrained violation metric

Figure 5: Supplementary results for abruptly changing environments.

In our scenario, we have a series of imitation learning tasks that are revealed sequentially. In each round $t$, a new task of Problem (55) is revealed, and its data $\mathcal{D}_t^{tr} = \{t_n, \hat{\mu}_n, \hat{\Sigma}_n\}_{n=1}^N$ and the collision area denoted by $\{g_i\}_{i=1}^m$ and $\{c_i\}_{i=1}^m$ are given. Then we apply the framework of online constrained meta-learning [Xu and Zhu, 2023] and construct the meta objective at round $t$ as

$$\mathcal{L}(\mathcal{A}lg(\lambda, \phi, \mathcal{D}_t^{tr}), \mathcal{D}_t^{val}) \tag{56}$$

with

$$\mathcal{A}lg(\lambda, \phi, \mathcal{D}_t^{tr}) = \underset{w}{\arg\min}\, \mathcal{L}(w, \mathcal{D}_t^{tr}) + \frac{\lambda}{2}\|w - \phi\|^2$$

$$\text{s.t.} \quad \frac{1}{N}\sum_{n=[N]} g_i(\xi(w, t_n)) \leq c_i,\ i = 1, \ldots, m, \tag{57}$$

where $\mathcal{L}(w, \mathcal{D}^{tr})$ is the loss function of the model parameter $w$ on a dataset $\mathcal{D}^{tr} = \{t_n, \hat{\mu}_n, \hat{\Sigma}_n\}_{n=1}^N$, and

$$\mathcal{L}(w, \mathcal{D}^{tr}) = \frac{1}{N}\sum_{n=1}^N \frac{1}{2}\left(\xi(w, t_n) - \hat{\mu}_n\right)^\top \hat{\Sigma}_n^{-1}\left(\xi(w, t_n) - \hat{\mu}_n\right). \tag{58}$$

Next, we update $\phi_t$ by our algorithms and compute $w_t = \mathcal{A}lg(\lambda, \phi_t, \mathcal{D}_t^{tr})$ as the task-specific model.

**General setup.** We use the demonstration data given by Huang et al. [2019]. In each round, the robot needs to imitate the given demonstration and write a capital letter. The demonstration dataset, denoted as $\mathcal{D}_t^{tr} = \{t_n, \hat{\mu}_n, \hat{\Sigma}_n\}_{n=1}^N$, consists of the sizes, angles, and locations of the letter. These sizes, angles, and locations are randomly sampled from a Gaussian distribution. The robot also needs to avoid a circle collision area, which defines $m$, $g_1$, and $c_1$ in (55) as $m = 1$, $c_1 = 0.5$ and $g_1(x) = \sigma(-\sqrt{(x_1 - d_1)^2 + (x_2 - d_2)^2} + r)$, where $\sigma(x) = \beta\log(1 + \exp(\beta x))$ is a barrier function and, $r$ is the radius of the collision area and $r = 2$. The center of the collision area, denoted by $(d_1, d_2)$, is sampled from the Gaussian distribution $\mathcal{N}(0, 1)$ for each task. Furthermore, in each task, the robot can access the full shot of demonstrations including 400 data points.

We model the position of the robot by a four-layer neural network with 128 which consists of an input layer of size 8, followed by 3 hidden layers of size 128 with the ReLU nonlinearities and an output layer of size 2. The neural network takes $\{t, t^2, t^3, t^4, sin(t), cos(2t), sin(2t), cos(2t)\}$ as the inputs and $q(w, t)$ as the outputs. We use the Adam optimizer [Kingma and Ba, 2014] with a learning rate of 0.001 for the optimization.

We simulate two types of dynamic environments: abruptly changing environments and gradually evolving environments. For abruptly changing environments, we use the demonstration data of capital letters "M", "E", "T" and "A" for the 4 stages. For gradually evolving environments, we rotate the letter "A" at a small random angle $\delta_t \in (0, 0.05]$ in each round. Specifically, at round $t$, we compute

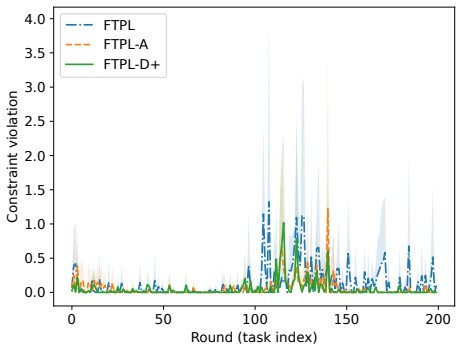
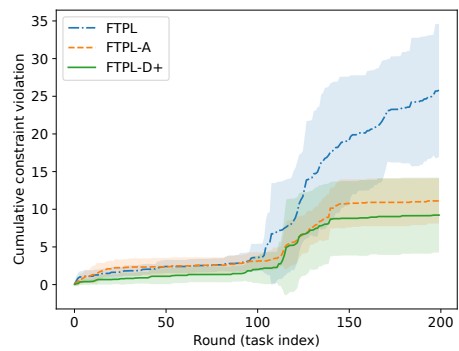

(a) Constrained violation metric    (b) Cumulative constrained violation metric

Figure 6: Supplementary results for gradually evolving environments.

the rotation angle $\delta = \sum_{\tau=1}^{t} \delta_\tau$, define a rotation matrix

$$R_\delta = \begin{bmatrix} \cos\theta & -\sin\theta \\ \sin\theta & \cos\theta \end{bmatrix},$$

and then transform $\hat{\mu}_n$ and $\hat{\Sigma}_n$ of the dataset by

$$\begin{cases} \hat{\mu}'_n = & \hat{\mu}_n R_\delta \\ \hat{\Sigma}'_n = & R_\delta^T \hat{\Sigma}_n R_\delta \end{cases}.$$

**Supplementary results.**    Apart from the loss shown in Fig 3, the constrained violation is another metric considered in Xu and Zhu [2023]. We plot the constrained violation in Fig 5 and Fig 6. The results show that our methods sustain the same and even less constrained violation than FTPL. This is reasonable and can be explained by Proposition 5 in Xu and Zhu [2023] that the constrained violation bound holds for any meta-parameter sequence $\phi_{1:T} = \{\phi_1, \cdots, \phi_T\}$.

## C.2    Experiments on few-shot image classification with robustness

**Problem formulation.**    Following the problem formulation in Chamon and Ribeiro [2020] and Xu and Zhu [2023], the problem of robust learning for a single image classification task $\mathcal{T}_t$ can be written as

$$\theta_t^* = \underset{\theta \in \Theta}{\operatorname{argmin}} \, \mathbb{E}_{z \sim \mathcal{D}_t} [\ell(\theta, z)] \tag{59}$$
$$\text{s.t.} \quad \mathbb{E}_{z \sim \mathcal{D}_{t,[P]}} [\ell(\theta, z)] - (1+\alpha)\mathbb{E}_{z \sim \mathcal{D}_t} [\ell(\theta, z)] \leq 0,$$

where $0 < \alpha < 1$ is the robustness tolerance parameter, $\ell$ is the loss function, $\mathcal{D}_t$ is the distribution of the original data, and $\mathcal{D}_{t,[P]}$ is the distribution of the perturbed data, which is generated by the PGD method on $\mathcal{D}_t$.

In our setting, a series of robust 5-way 5-shot image classification tasks are revealed sequentially. In each round $t$, a new task of Problem (59) is revealed, which consists of a 5-shot meta-training dataset $\mathcal{D}_t^{tr}$ (the support dataset), and a meta-validation dataset $\mathcal{D}_t^{val}$ (the query dataset). Additionally, the dataset $\mathcal{D}_{t,[P]}^{tr}$ is also 5-shot and generated by the PGD method [Kurakin et al., 2018] on $\mathcal{D}_t^{tr}$. Then the meta objective at round $t$ can be written as

$$\mathcal{L}(\mathcal{A}lg(\lambda, \phi, \mathcal{D}_t^{tr}), \mathcal{D}_t^{val}) \tag{60}$$

with

$$\mathcal{A}lg(\lambda, \phi, \mathcal{D}_t^{tr}) = \underset{\theta}{\arg\min} \, \mathcal{L}(\theta, \mathcal{D}_t^{tr}) + \frac{\lambda}{2}\|\theta - \phi\|^2 \tag{61}$$
$$\text{s.t.} \quad \mathcal{L}(\theta, \mathcal{D}_{t,[P]}^{tr}) - (1+\alpha)\mathcal{L}(\theta, \mathcal{D}_t^{tr}) \leq 0,$$

where $\mathcal{L}(w, \mathcal{D}_t^{tr})$ is the loss function of the model parameter $\theta$ on a dataset $\mathcal{D}_t^{tr}$. Next, we update $\phi_t$ by our algorithms and compute $\theta_t = \mathcal{A}lg(\lambda, \phi_t, \mathcal{D}_t^{tr})$ as the task-specific model. In the experiment, we select $\alpha = 0.3$ and $\lambda = 1.0$.

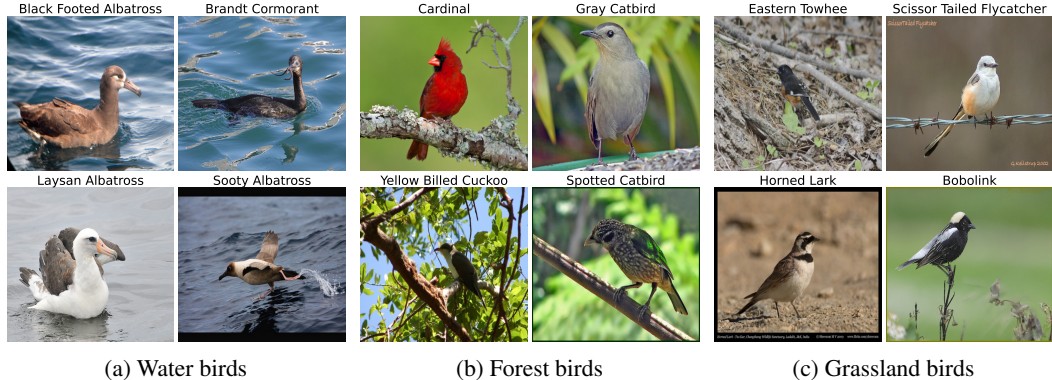

|  |  |  |
|:---:|:---:|:---:|
| (a) Water birds | (b) Forest birds | (c) Grassland birds |

Figure 7: Example images of birds from different habitats.

**General setup.** We conduct the experiments on CUB-200-2011 (referred to as CUB) dataset [Wah et al., 2011]. The input image processing and network architecture are fully provided in Section 6.2. To simulate real-world dynamic environments, we select three groups of bird categories from the CUB dataset based on their habitats: water birds, forest birds, and grassland birds. We show example images in Fig. 7 and provide a full list of classes for each group in Table 2. For each group, we randomly allocate 50% of the classes for training data, 25% for validation data, and 25% for test data. The time horizon $T$ is then evenly divided into three stages, with tasks sampled from one group of bird categories in each stage. Specifically, in each round, we sample a 5-way 5-shot task from the training data classes, involving the selection of 5 classes with 5 images per class. During the meta-testing phase, we sampled 600 5-way 5-shot tasks from the test data classes.

We use the Adam optimizer [Kingma and Ba, 2014] with a learning rate of 0.001 for the optimization and use cross-entropy as the loss function. The adversarial attack on the query set is performed using the PGD method with a perturbation size of $\epsilon = 2/255$. This process involves 7 iterative steps, each with a step size of $2.5\epsilon$.

Table 2: Bird species by habitat type in the CUB dataset.

| Habitat Type | CUB Label Names |
| --- | --- |
| **Water** | 001.Black_footed_Albatross, 002.Laysan_Albatross, 003.Sooty_Albatross, 023.Brandt_Cormorant, 024.Red_faced_Cormorant, 025.Pelagic_Cormorant, 050.Eared_Grebe, 051.Horned_Grebe, 052.Pied_billed_Grebe, 053.Western_Grebe, 058.Pigeon_Guillemot, 059.California_Gull, 060.Glaucous_winged_Gull, 061.Heermann_Gull, 062.Herring_Gull, 064.Ring_billed_Gull, 065.Slaty_backed_Gull, 066.Western_Gull, 071.Long_tailed_Jaeger, 072.Pomarine_Jaeger, 084.Red_legged_Kittiwake, 086.Pacific_Loon, 089.Hooded_Merganser, 090.Red_breasted_Merganser, 100.Common_Yellowthroat, 101.White_Pelican, 106.Horned_Puffin |
| **Forest** | 017.Cardinal, 018.Spotted_Catbird, 019.Gray_Catbird, 033.Yellow_billed_Cuckoo, 035.Purple_Finch, 038.Great_Crested_Flycatcher, 054.Blue_Grosbeak, 057.Rose_breasted_Grosbeak, 067.Anna_Hummingbird, 068.Ruby_throated_Hummingbird, 070.Green_Violetear, 073.Blue_Jay, 075.Green_Jay, 095.Baltimore_Oriole, 096.Hooded_Oriole, 103.Sayornis, 111.Loggerhead_Shrike, 152.Blue_headed_Vireo, 154.Red_eyed_Vireo, 158.Bay_breasted_Warbler, 160.Black_throated_Blue_Warbler, 162.Canada_Warbler, 166.Golden_winged_Warbler, 175.Palm_Warbler, 183.Northern_Waterthrush, 185.Bohemian_Waxwing, 188.Pileated_Woodpecker, 193.Carolina_Wren, 195.House_Wren, 200.Common_Yellowthroat |
| **Grassland** | 009.Brewer_Blackbird, 010.Red_winged_Blackbird, 013.Bobolink, 021.Eastern_Towhee, 041.Scissor_tailed_Flycatcher, 085.Horned_Lark, 104.American_Pipit, 110.Geococcyx, 113.Baird_Sparrow, 114.Black_throated_Sparrow, 115.Brewer_Sparrow, 116.Chipping_Sparrow, 121.Grasshopper_Sparrow, 127.Savannah_Sparrow, 128.Seaside_Sparrow, 129.Song_Sparrow, 131.Vesper_Sparrow, 140.Scarlet_Tanager, 146.Forsters_Tern, 149.Brown_Thrasher, 150.Sage_Thrasher |

