# OpenReview forum: "Online Non-convex Learning in Dynamic Environments"
_NeurIPS.cc/2024/Conference — NeurIPS 2024 poster_

### Official Review · Reviewer_wXns · 2024-06-28

**Soundness:** 4
**Presentation:** 3
**Contribution:** 2
**Rating:** 6
**Confidence:** 3

**Summary:**

This paper provides $O(V_T^{1/3}T^{2/3})$ dynamic regret
and $O(\sqrt{\tau\log(T)})$ strongly-adaptive regret guarantees for
online learning with Lipschitz losses in bounded domains.

**Strengths:**

The paper fills a gap in the literature by providing the
dynamic and strongly-adaptive extensions of the FTPL
algorithm. The paper reads well and is easy to follow.

**Weaknesses:**

The results are straight-forward applications of existing results. Prior analyses provide the
regret bound of FTPL, and by assuming Lipschitz+bounded domain one immediately has access
to an experts algorithm which will combine instances of the base algorithm (multiplicative weights update).
Similarly, the strongly adaptive result seems to follow easily from the standard
geometric covering intervals + AdaNormalHedge.

The results are adaptive to the temporal variability, $\sum_{t=2}^{T}\sup_x|f_t(x)-f_{t-1}(x)|$, which is more
of a variance measurement than it is a measure of comparator variability. For instance, even
the temporal variability can be high even if the comparator is fixed.
Yet studying dynamic
regret in the first place implies we are concerned with the variability of the comparator sequence; the ideal measure
is something like path-length, where the "difficulty" of the comparator sequence is directly
reflected in the bound. Here, the bound reflects only seems to
reflect the difficulty of the loss sequence.

**Questions:**

- what are some examples of interesting losses which are non-convex, but bounded and lipschitz?
- Can a similar strategy be developed for FTRL? Or is randomization an essential ingredient for the base algorithm?
- Most online learning guarantees are anytime and wp1, while the results here are only in expectation.
  Would it be difficult to get a stronger high-probability result, at least?

**Limitations:**

The weakness in terms of adapting to temporal variability instead of path-length is pointed out in the conclusion

---

> ### Author Rebuttal · Authors · 2024-08-05
>
> Many thanks for the constructive reviews!
>
> ---
>
> **Q1**: What are some examples of interesting losses which are non-convex, but bounded and lipschitz?
>
> **A1**: An illustrative example is the loss function of Generative Adversarial Networks (GANs), which has been extensively discussed by Agarwal et al. [2019, Section 4.1].
>
> ---
>
> **Q2**: Can a similar strategy be developed for FTRL? Or is randomization an essential ingredient for the base algorithm?
>
> **A2**: We believe that FTRL cannot achieve similar results, as Proposition 3 of Suggala and Netrapalli [2020] demonstrates that deterministic algorithms cannot attain sublinear regret in the context of online non-convex learning. Therefore, it is essential to employ randomization in base algorithms.
>
> ---
>
> **Q3**: Most online learning guarantees are anytime and wp1, while the results here are only in expectation. Would it be difficult to get a stronger high-probability result, at least?
>
> **A3**: Thank you for the valuable feedback. It is indeed feasible to establish high-probability regret bounds. We can adapt the approach used in the proof of Lemma 4.1 by Cesa-Bianchi and Lugosi [2006], particularly their technique for deriving high-probability bounds. Given our assumptions, the discrepancy between the actual loss and the expected loss in each round is bounded. Consequently, we can apply the Hoeffding-Azuma inequality for martingale differences, and extend the expected regret bound to a high-probability one.

---

> > ### Comment · Reviewer_wXns · 2024-08-07
> >
> > Thank you for the detailed response!
> >
> > Sorry I missed this in my original review, but could you briefly point to which techniques or steps in the analysis can *not* be found in existing works? I am still having trouble understanding to what extent the analysis and techniques are novel. Skimming through the appendix, nothing has jumped out at me as being particularly surprising or new. It is of course possible to derive novel results purely using existing techniques; if that's the case, that is fine.

---

> > > ### Author Response · Authors · 2024-08-08
> > >
> > > Dear Reviewer wXns,
> > >
> > > Thank you for your question. Compared to existing online algorithms for dynamic environments, our main distinction lies in the utilization of randomized sampling in the process of tracking experts to handle non-convexity. This approach aligns closely with the work of Suggala and Netrapalli [2020], which shows that with the help of randomization, FTPL can achieve optimal (static) regret for non-convex losses.
> > >
> > > On the other hand, we would also like to emphasize that the integration of multiple techniques to address the challenges of non-stationarity and non-convexity in our algorithm is a significant undertaking. These techniques include the restarting strategy [Besbes et al., 2015], the two-layer meta-expert framework for parameter selection [van Erven and Koolen, 2016], randomization [Cesa-Bianchi and Lugosi, 2006], geometric covering [Daniely et al., 2015], sleeping experts [Luo and Schapire, 2015], and the reduction from dynamic regret to strongly adaptive regret [Zhang et al., 2018b]. This comprehensive integration is far from trivial; it requires a deep understanding of various theoretical concepts and the ability to apply them effectively in practice.
> > >
> > > Best
> > >
> > > Authors

---

> > > > ### Comment · Reviewer_wXns · 2024-08-08
> > > >
> > > > Great, thanks; after considering the responses to my and the other reviewer's concerns, I've decided to raise my score.

---

> > > > > ### Author Response · Authors · 2024-08-08
> > > > >
> > > > > Dear Reviewer wXns,
> > > > >
> > > > > Thank you for your kind reply! We will enhance our paper based on your insightful comments.
> > > > >
> > > > >
> > > > > Best
> > > > >
> > > > > Authors

---

### Official Review · Reviewer_njVm · 2024-06-30

**Soundness:** 2
**Presentation:** 3
**Contribution:** 2
**Rating:** 5
**Confidence:** 3

**Summary:**

This paper presents novel algorithms to tackle the challenges of online learning with non-convex loss functions in dynamic settings. The authors extend the Follow the Perturbed Leader (FTPL) algorithm to dynamic environments by proposing two new algorithms: FTPL-D and FTPL-D+. They demonstrate the effectiveness of their methods through theoretical regret analysis.

**Strengths:**

The paper introduces innovative extensions to the FTPL algorithm, specifically designed for non-convex and dynamic environments. The paper is well-written and structured, making it easy to follow the theoretical developments and experimental setups. The algorithms are clearly described, and the results are presented in a manner that highlights their significance.

**Weaknesses:**

I think the problem addressed in this paper is not sufficiently central. The main contribution is extending the loss function to a non-convex case under this specific setup, and the proof methods are quite similar to previous works. Additionally, the experimental setup is relatively simple and lacks more practical experiments.

**Questions:**

Could the authors include some more challenging experimental tasks? The current tasks are relatively simple.

**Limitations:**

The authors did not discuss the limitations in sufficient detail. The last paragraph of the paper only contains some discussion on future directions.

---

> ### Author Rebuttal · Authors · 2024-08-05
>
> Many thanks for the constructive reviews!
>
> We will add more experiments in the final version. Here, we want to highlight that although each imitation task in our experimental setup is relatively simple, the framework of online constrained meta-learning introduces considerable complexity, especially in dynamic environments. We elaborate on this in two aspects.
>
> 1. It is important to note that the demonstration trajectories to be imitated are collected in free space, without any knowledge of obstacle avoidance, whereas the task is performed in a new cluttered environment. Before the task is revealed, the obstacles in the new environment are unknown to the learner. Therefore, the learner has to quickly adapt to the presence of obstacles.
>
> 2. Additionally, in the two types of dynamic environments we simulated, the changes in task trajectories are also unknown. This implies that the learner needs to adapt to these changes in trajectories to better imitate the current task's demonstration.

---

> > ### Comment · Reviewer_njVm · 2024-08-13
> > **Thank you**
> >
> > Thank you very much for your response. I did not see any additional experiments provided, so I will maintain my current score.

---

> > > ### Author Response · Authors · 2024-08-14
> > >
> > > Dear Reviewer njVm,
> > >
> > > Thank you for your kind reply! We are currently conducting additional experiments and will include more results in the final version of the paper.
> > >
> > > Best
> > >
> > > Authors

---

### Official Review · Reviewer_PVQP · 2024-07-01

**Soundness:** 3
**Presentation:** 4
**Contribution:** 3
**Rating:** 6
**Confidence:** 4

**Summary:**

The paper proposes two variant algorithms of the FTPL (follow-the-perturbed-leader) algorithm for online learning with non-convex losses in time-changing environments. The authors analyze these algorithms under the settings where the variability of the environment ($V_T$) is both known and unknown. FTPL-D is their proposed algorithm to minimize dynamic regret, and FTPL-A is their proposed algorithm to minimize adaptive regret, where their regret bounds match the best known bounds in the convex equivalent cases. The authors also provide an imitation-learning-based example to test the validity of their algorithms.

**Strengths:**

The paper introduces (to the best of my knowledge) novel _variants_ of the FTPL algorithm. The paper is well-organized and easy to follow, with clear explanations of the theoretical concepts and practical implementation details. The authors provide detailed proofs of their theoretical results in the appendices, as well as a numerical experiment to empirically document the performance of the algorithm in a concrete example setting to show that their regret bounds are competitive. The chief technical contribution (perhaps, surprisingly, to me) is that the non-convex adaptive-regret/dynamic-regret bounds match their convex equivalents, where the distinction follows mainly from the oracle requirements.

**Weaknesses:**

1) Going through the proofs: in equation 17 (to bound $b_i$, it seems that the last inequality is very loose. You are bounding two differences by two summations of differences. I wonder if this can be strengthened?
2) The analysis for these algorithms seem to be quite straightforward - most of the proofs follow very standard analytic methods in the literature. I wonder why this result has not been discovered earlier?
3) Is it realistic to have an $O(1/\sqrt{\gamma}, 1/\gamma)$-approximate oracle? In particular, I was under the impression that the oracle defined in equation 5 was for a constant value of the parameter $\gamma$, but it seems that the choice of $\gamma$ might depend on $L, d, D$, and $T$ (which can be asymptotically large). Doesn't this also make the offline optimization oracle intractable?
4) If the dynamic environment changes very quickly, how well can the system practically handle this? i.e. if the variational intensity $V_T$ is very high, the experiments do not seem to model this situation at all...
5) Further, the computational complexity of this algorithm does not seem well-analyzed. What is the memory requirement of each algorithm? How does it scale with T and with the number of experts?
6) One minor suggestion is that the authors should compare these proposed algorithms with SOTA algorithms (like online gradient descent (OGD) and its oracle). Further, there appears to be a line of related recent works from the control theory + online optimization literature such as [a,b,c,d] which has also made progress in this problem. For instance, [c] appears to have a more tractable oracle, in particular, than OGD. I wonder if you can also compare the analytic methods and results to these works as well, if they are in fact related?

References:

[a] Online Policy Optimization in Unknown Nonlinear Systems [Lin et. al. 2024]

[b] Adaptive Regret for Control of Time-Varying Dynamics [Gradu et. al. 2020]

[c] Online Adaptive Policy Selection in Time-Varying Systems: No-Regret via Contractive Perturbations [Lin et. al. 2024]

[d] Online Control of Unknown Time-Varying Dynamical Systems [Minasyan et. al. 2022]

**Questions:**

Please address the questions from the above section.

**Limitations:**

Not applicable.

---

> ### Author Rebuttal · Authors · 2024-08-04
>
> Many thanks for the constructive reviews!
>
> ---
> **Q1**: Going through the proofs: in equation 17 ... if this can be strengthened?
>
> **A1**: In our opinion, it seems impossible to strengthen this proof. The reason is that our analysis of $b_i$ aligns with that used in the convex case, and the derived $O(T^\frac{2}{3}(V_T+1)^\frac{1}{3})$ dynamic regret bound is minimax optimal [Besbes et al., 2015], and thus unimprovable.
>
> ---
>
> **Q2**: The analysis for these algorithms seem … why this result has not been discovered earlier?
>
> **A2**: The reason may be that previous work on dynamic regret and adaptive regret has primarily focused on convex functions, often considering non-convex optimization too challenging to yield meaningful results. Recently, we were inspired by Suggala and Netrapalli [2020], who demonstrated that online non-convex optimization could also achieve optimal regret, prompting us to undertake this work.
>
> ---
>
> **Q3**: Is it realistic to have an $\mathcal{O}(1/\sqrt\gamma,1/\gamma)$-approximate oracle? ...
>
> **A3**: We are sorry for the confusion, and clarify this issue below.
> 1. Recall that $\alpha$ and $\beta$ are parameters controlling the precision of the optimization oracle, with their values adjustable to achieve specific regret bounds (of course, smaller values typically lead to higher computational costs). In our submitted manuscript, we preset the values of $\alpha$ and $\beta$ solely to streamline the presentations. Actually, when analyzing static regret, Suggala and Netrapalli [2020, Page 4] also simplify their bound by setting $\alpha = O(1/\sqrt{T})$ and $\beta = O(1/T)$.
>
> 2. Following the suggestion of reviewers, we will revise our theorems to explicitly incorporate $\alpha$ and $\beta$, as detailed in our **global response** to all reviewers. For example, **Theorem 1** can be rewritten as: under Assumptions 1 and 2, and setting $\eta = 1/\sqrt{d\gamma}$, Algorithm 2 ensures
> $$
> \begin{equation}
>     \mathbb{E}\left [  R_D^* \right ]
>     \le O\left (\frac{(1+\alpha \sqrt \gamma+ \beta \gamma)T}{\sqrt{\gamma }}  + \gamma V_T \right) .
> \end{equation}
> $$ If the value of $V_T$ is known, we set $\gamma = \min \left \\{\left\lfloor (\frac{T}{V_T})^\frac{2}{3}\right\rfloor, T \right \\}$, then we have
> $$
> \begin{align}
>     \mathbb{E}\left [ R_D^* \right ] &\le O((1+ \alpha \sqrt T+\beta T )T^\frac{2}{3}(V_T+1)^\frac{1}{3} ).\nonumber
> \end{align}
> $$
> It can be inferred that when $\alpha = O(1/\sqrt{T})$ and $\beta = O(1/T)$, which mirrors the settings used by Suggala and Netrapalli [2020, Page 4], Algorithm 2 achieves an $O(T^\frac{2}{3}(V_T+1)^\frac{1}{3})$ dynamic regret bound.
> ---
>
> **Q4**: If the dynamic environment changes very quickly, how well ...
>
> **A4**: If the environment changes quickly and $V_T$ is very high, the system's performance will decline. That is because our dynamic regret bound of $O(T^\frac{2}{3}(V_T+1)^\frac{1}{3})$ is minimax optimal  [Besbes et al., 2015]. An increase in $V_T$ will lead to a larger lower bound for dynamic regret, which implies that the system's loss will increase.
>
> ---
>
> **Q5**: Further, the computational complexity of this algorithm does not seem well-analyzed ...
>
> **A5**: Below we will discuss the computational complexity and the memory requirement of our algorithms.
>
> * ***Computational Complexity***: It is important to note that the computational bottleneck of our algorithms lies in the offline optimization oracle. Therefore, we can simply define the computational complexity of the algorithms in terms of the number of oracle calls. For Algorithm 2, only one oracle call is needed per round. For Algorithm 3, there are $ N = \left \lfloor \log T \right \rfloor$ experts in each round, with each expert requiring one oracle call, so the number of oracle calls per round is $O(\log T)$. Similarly, for Algorithm 4, in the $t$-th round, there are $N_t = 1+\left \lfloor \log t \right \rfloor $ experts, thus requiring $O(\log T)$ oracle calls per round.
> * ***Memory requirement***: First, in order to run the optimization oracle, we need to store all the functions in memory, which takes $O(t)$ space in the $t$-th round. Thus, Algorithm 2 has $O(t)$ space complexity, which is the same as Suggala and Netrapalli [2020]. Second, each active expert consumes a constant memory. According to the discussion above, Algorithms 3 and 4 require $O(t + \log T)$ space, since they need to maintain $O(\log T)$ experts.
> ---
>
> **Q6**: the authors should compare these proposed algorithms with SOTA algorithms ...
>
> **A6**: Thanks for the suggestion. We compare our algorithms with OGD and the methods in [a, b, c, d] from the following aspects:
>
> * ***Oracle***: Our algorithm relies on an offline optimization oracle, whereas OGD uses a gradient oracle, which returns the gradient information of the loss function at the decision point. Additionally, we note that [c] introduced a more tractable oracle, which approximates the true gradient by computing the gradient on the actual trajectory, thereby reducing the computational complexity. Generally, a gradient oracle is easier to obtain and has lower computational complexity compared to an offline optimization oracle.
>
> * ***Metrics and results***: In dynamic environments, OGD achieves: (i) minimax optimal dynamic regret of $\sqrt{T(P_T+1)}$ for general convex functions [Zhang et al., 2018a], using path-length $P_T$ (defined in Appendix B) to measure environmental variation; (ii) local regret of $\sqrt{T(V_T+1)}$ for non-convex functions [c], also using $V_T$ to measure environmental variation.  In [a, c], the performance of the algorithms is measured by _local regret_ [Hazan et al., 2017], which focuses on tracking changing stationary points. In contrast, we use _regret_ (including dynamic regret and strongly adaptive regret), which is a stronger metric. [b] focuses on adaptive regret, and [d] on strongly adaptive regret, but both assume general convex cost functions, whereas our paper supports non-convex functions.

---

> > ### Comment · Reviewer_PVQP · 2024-08-09
> > **Clarifications**
> >
> > I thank the authors for their detailed reply.
> >
> > Regarding the memory requirement, if the algorithm needs O(t) space in the t'th round, then this space requirements increase with the length of the game... Why does this not render the algorithm intractable?

---

> > > ### Author Response · Authors · 2024-08-09
> > >
> > > Dear Reviewer PVQP,
> > >
> > > The reason lies in the fact that online non-convex learning is highly challenging, leading to a general acceptance of some compromises in computational or space complexity. Notably, even with an $O(t)$ space complexity, the work of Suggala and Netrapalli [2020] received the Best Student Paper Award at ALT 2020, indicating that the online learning community is open to embracing imperfect algorithms. Moreover, given the current state of research in online non-convex learning, the offline optimization oracle is the most acceptable assumption, compared to the intractable sampling oracle.
> > >
> > > In practice, we can sample functions and store a subset of them to avoid $O(t)$ storage space requirements. Alternatively, we can employ continual learning techniques to incrementally implement an offline optimization oracle, which typically does not require storing all functions. Thus, we believe our results hold theoretical value and can guide the design and implementation of practical algorithms.
> > >
> > > Best
> > >
> > > Authors

---

> > > > ### Comment · Reviewer_PVQP · 2024-08-11
> > > > **More clarifications**
> > > >
> > > > I thank the authors for their detailed reply.
> > > > Re: the space complexity, this makes sense.
> > > >
> > > > One other (tangential) question regarding my earlier point on tightening equation 17, I wonder if it might be possible to find certain instances where equation 17 could be tightened? I'm not very familiar with the lower bound - is it an unconditional/instance-independent lower bound? If not, it might be possible to find "easy instances" of this problem that can be solved with a possibly lower regret I wonder?
> > > >
> > > > Thank you, and I will retain my original score for now.

---

> ### Author Response · Authors · 2024-08-12
>
> Dear Reviewer PVQP,
>
> Thank you for your kind reply! We will revise our paper based on your constructive comments.
>
> Regarding your question, the lower bound is not an unconditional one, so it is indeed possible to find some "easy instances" that yield a better bound. And we plan to investigate this issue further in the future. However, at present, we believe this is quite challenging. Although "bounding two differences by two summations of differences" might appear loose, it is actually *tight* in the worst case. For example, when the function changes only once within the interval $[q_i, q_{i+1} -1]$, i.e.,
> $$
> f_{q_i-1} (\cdot )  =f_{q_i} (\cdot ) =\cdots = f_{t-1}(\cdot) \neq f_{t}(\cdot) = \cdots f_{ q_{i+1} -1} (\cdot ),
> $$
> we have
> $$
> V_T(i)= \sum_{k=q_i} ^{q_{i+1}-1} \max_{\mathbf{x} \in \mathcal{K}} |f_k(\mathbf{x})-f_{k-1}(\mathbf{x})| =  \max_{\mathbf{x} \in \mathcal{K}} |f_t(\mathbf{x})-f_{t-1}(\mathbf{x})|=  \max_{\mathbf{x} \in \mathcal{K}} |f_t(\mathbf{x})-f_{q _i}(\mathbf{x})|
> $$
> which, in the worst case, could equal
> $$
> f_t(\mathbf{x} _{q_i}^*) -  f _{q _i}  (\mathbf{x} _{q_i}^*).
> $$
>
>
> Best
>
> Authors

---

> ### Comment · Reviewer_PVQP · 2024-08-12
> **More clarifications**
>
> I thank the authors for their reply. I think finding instance-dependent bounds would be an exciting area of future study - but agree now that for the general case, this inequality is tight.
>
> Another question that comes to mind is whether the oracle assumption can be weakened in any form? I understand the work from Suggala and Netrapalli (2010) also assumes the existence of this strong oracle, but is this "strong oracle assumption" critical to the main result? For instance, could there be a tradeoff between "many weak (constant parameter) stochastic oracles" versus "one strong oracle"?
>
> Thanks!

---

> > ### Author Response · Authors · 2024-08-13
> >
> > [It seems that our previous response did not trigger an email notification, so we are resending our reply]
> >
> > Dear Reviewer PVQP,
> >
> > Thank you for your response. From our understanding, the "strong oracle assumption" is crucial for achieving optimal global regret. Indeed, some works in the literature, such as Guan et al. [2023], employ weaker oracle assumptions, but they can only provide theoretical guarantees for local regret.
> >
> > Additionally, it's worth mentioning that although our algorithm requires calling the optimization oracle $O(\log T)$ times per round, these calls do not need to have the same precision. For example, in Algorithm 4, the precision of the oracle is related to the length of the interval in Figure 2. For shorter intervals, we can use a lower-precision oracle to reduce computational cost. However, for longer intervals, a high-precision oracle is necessary.
> >
> >
> > Best
> >
> > Authors

---

> ### Author Response · Authors · 2024-08-13
>
> Dear Reviewer PVQP,
>
> Thank you for your response. From our understanding, the "strong oracle assumption" is crucial for achieving optimal global regret. Indeed, some works in the literature, such as Guan et al. [2023], employ weaker oracle assumptions, but they can only provide theoretical guarantees for local regret.
>
> Additionally, it's worth mentioning that although our algorithm requires calling the optimization oracle $O(\log T)$ times per round, these calls do not need to have the same precision. For example, in Algorithm 4, the precision of the oracle is related to the length of the interval in Figure 2. For shorter intervals, we can use a lower-precision oracle to reduce computational cost. However, for longer intervals, a high-precision oracle is necessary.
>
>
> Best
>
> Authors

---

### Official Review · Reviewer_fFaA · 2024-07-11

**Soundness:** 2
**Presentation:** 3
**Contribution:** 2
**Rating:** 5
**Confidence:** 2

**Summary:**

This paper considers the problem of non-convex online learning in dynamic environments. The authors go on to provide some algorıthmic variants and their respective regret bounds depending on the functional variation. They show that the non-convex dynamic regrets matches the convex ones in the literature.

**Strengths:**

S1) Regret analysis for non-convex online learning is highly significant.

S2) Deriving dynamic regret bounds for non-convex optimization with existing tools for convex optimization is nice.

S3) Achieving dynamic regret and adaptive regret simultanously is powerful.

**Weaknesses:**

W1) While the derivations are nice, algorithmic contribution is limited.

W2) Existance of an approximate optimization oracle with selectable parameters seems dubious.

W3) Since the $(\alpha,\beta)$-approximate oracle is an integral part of the algorithm, providing regret bounds independent of $(\alpha,\beta)$ seems counter-intuitive.

**Questions:**

Major Question:

My biggest issue is with the missing feasibility discussion. The existence of an $(\alpha,\beta)$-approximate oracle is assumed, which is not a problem by itself. However, in the derivations, $(\alpha,\beta)$ can be set freely depending on $\gamma$, which is an input. This seems like a big issue to me. For static regret, the effect of $\gamma$ is non-existing and regret results may naturally depend on given $\alpha, \beta$, which is not the case for your analyses. Given that all of the results depend on this, I would like for authors to properly address it. Am I missing something here?

\
Minor Questions:

- Line 152: what is $x^*$?

- Line 179: how do you set $\gamma$?

**Limitations:**

Most limitations are adequately addressed except for the existence of a parameter-dependent offline optimization oracle.

---

> ### Author Rebuttal · Authors · 2024-08-04
>
> Many thanks for the constructive reviews! Detailed responses have been provided for all the questions raised. In particular, we believe that the major question can be fully addressed by restating our theorems. We hope the reviewer could examine them, and re-evaluate our paper. We are looking forward to addressing any further questions in the author-reviewer discussion period.
>
>
> ---
> **Q1**: My biggest issue is with the missing feasibility discussion ...
>
> **A1**: We are sorry for the confusion, and clarify this issue below.
> 1. Recall that $\alpha$ and $\beta$ are parameters controlling the precision of the optimization oracle, with their values adjustable to achieve specific regret bounds (of course, smaller values typically lead to higher computational costs). In our submitted manuscript, we preset the values of $\alpha$ and $\beta$ solely to streamline the presentations. Actually, when analyzing static regret, Suggala and Netrapalli [2020, Page 4] also simplify their bound by setting $\alpha = O(1/\sqrt{T})$ and $\beta = O(1/T)$.
>
> 2. Following the suggestion of reviewers, we will revise our theorems to explicitly incorporate $\alpha$ and $\beta$, as detailed in our **global response** to all reviewers. For example, **Theorem 1** can be rewritten as: under Assumptions 1 and 2, and setting $\eta = 1/\sqrt{d\gamma}$, Algorithm 2 ensures
> $$
> \begin{equation}
>     \mathbb{E}\left [  R_D^* \right ]
>     \le O\left (\frac{(1+\alpha \sqrt \gamma+ \beta \gamma)T}{\sqrt{\gamma }}  + \gamma V_T \right) .
> \end{equation}
> $$ If the value of $V_T$ is known, we set $\gamma = \min \left \\{\left\lfloor (\frac{T}{V_T})^\frac{2}{3}\right\rfloor, T \right \\}$, then we have
> $$
> \begin{align}
>     \mathbb{E}\left [ R_D^* \right ] &\le O((1+ \alpha \sqrt T+\beta T )T^\frac{2}{3}(V_T+1)^\frac{1}{3} ).\nonumber
> \end{align}
> $$
> It can be inferred that when $\alpha = O(1/\sqrt{T})$ and $\beta = O(1/T)$, Algorithm 2 achieves an $O(T^\frac{2}{3}(V_T+1)^\frac{1}{3})$ dynamic regret bound.
>
> ---
>
> **Q2**: Line 152: what is $x^*$?
>
> **A2**: We apologize for the typo in Line 152, where we mistakenly wrote $x \in \mathcal{K}$ instead of $x^* \in \mathcal{K}$. Here $x^*$ denotes the decision returned by the $(\alpha,\beta)$-approximate optimization oracle.
>
> ---
>
> **Q3**: Line 179: how do you set $\gamma$?
>
> **A3**: We set $\gamma$ according to the value of $V_T$, as shown in Appendix A.1. Specifically, if $V_T \ge \frac{1}{\sqrt{T}}$, we choose $\gamma = \left\lfloor (\frac{T}{V_T})^\frac{2}{3}\right\rfloor $; otherwise we choose $\gamma = T$.
>
> ---
>
> **Q4**: Existance of an approximate optimization oracle with selectable parameters seems dubious.
>
> **A4**: First, as explained in **A1**, we can reformulate our theorems to explicitly incorporate $\alpha$ and $\beta$. In this way, we obtain the regret bounds shown in the **global response**, treating $\alpha$ and $\beta$ as constants. Second, it is important to note that our algorithms are not restricted to using only the approximate optimization oracle. We have the flexibility to select any algorithm with a proven static regret guarantee in online non-convex learning as our subroutine, which may not depend on the approximate optimization oracle.
>
> ---
>
> **Q5**: While the derivations are nice, algorithmic contribution is limited.
>
> **A5**: We acknowledge that the techniques employed in this paper may not be entirely novel. However, the integration of these methods and the necessary adaptations we have made represent significant tasks. Importantly, this work is pioneering in showing that it is feasible to achieve optimal dynamic and adaptive regret in the context of online non-convex optimization when an approximate optimization oracle is available.

---

> ### Comment · Reviewer_fFaA · 2024-08-09
>
> I appreciate your rebuttal. I have read the other reviews and rebuttals as well. In all honesty, your response further confused me. With your proposed selection of $\alpha,\beta$, if we set $\sigma$ to all zero in Definition 1, the offline oracle is already $1/\sqrt{T}$ approximate. If we just use the selection of the oracle, the regret becomes $\sqrt{T}$. Note only is the order better than your result, it is also independent from $V_T$, which is weird. Am I missing something here?

---

> > ### Author Response · Authors · 2024-08-09
> >
> > Dear Reviewer fFaA,
> >
> > The reason is that we *cannot* set $\sigma$ to all zero, due to the non-convex nature of the problems. To achieve valid static regret, it is crucial to employ the strategy of follow the *perturbed* leader, meaning the elements of $\sigma$ must be sampled from an exponential distribution. Setting $\sigma=0$ would leave us without any theoretical guarantees for online non-convex learning.
> >
> > Additionally, we would like to clarify that our paper focuses on *dynamic regret* and *adaptive regret*. In particular, the dependence of our dynamic regret bounds on $V_T$ is optimal [Besbes et al., 2015]. Although the $O(\sqrt{T})$ bound of Suggala and Netrapalli [2020] is independent from $V_T$, it pertains only to static regret.
> >
> >
> > Best
> >
> > Authors

---

> > > ### Comment · Reviewer_fFaA · 2024-08-10
> > >
> > > Definition 1 is lacking in its current form. There exists no mention of the limitations/requirements. The issue is some selections of $\sigma$ make an infeasible task seem feasible. However, selection of $\sigma$ is mentioned only by its exponential distribution; and you do not have to choose $\eta$ to be infinity, even by selecting $\eta=\sqrt{T}$, same weird result pops up. In needs to be properly addressed in Definition 1, the criterion on $\alpha,\beta,\eta$; that allows you to set as input and also allows for the existence of an approximate minimizer.

---

> ### Author Response · Authors · 2024-08-10
>
> Dear Reviewer fFaA,
>
> Thank you for your reply. We will revise this part to enhance clarity.
>
> Our Definition 1 is the same as the one introduced by Suggala and Netrapalli [2020]. It essentially posits the existence of a powerful optimization oracle capable of returning an approximate solution to an optimization problem that combines a non-convex function $f(x)$ and a linear function $\langle \sigma, x \rangle$. This assumption was first made by Agarwal et al. [2019] and has since been widely utilized in the study of online non-convex optimization. Existing research typically accepts this assumption without challenging its validity.
>
> Assuming access to an offline optimization oracle is considered reasonable because straightforward algorithms, such as stochastic gradient descent, can quickly find approximate global optima, even for non-convex objective functions. It is important to note that, with such an oracle, online non-convex learning remains challenging. This is because the oracle is designed for offline optimization and cannot be directly used to bound regret.
>
> We are somewhat unclear about your concern. For instance, when you mention, "even by selecting $\eta=\sqrt{T}$, same weird result pops up," could you please clarify what you mean by "weird result"?
>
> Best
>
> Authors

---

> > ### Comment · Reviewer_fFaA · 2024-08-13
> >
> > What I mean is: it seems like with your choice of $\alpha, \beta$ and $\eta=\sqrt{T}$, the dynamic regret $O(\sqrt{T})$ is achievable just by utilizing the outputs of the approximate oracle. This seems weird. Nonetheless, I have read the other comments/rebuttals and changing my score/confidence accordingly.

---

> > > ### Author Response · Authors · 2024-08-14
> > >
> > > Dear Reviewer fFaA,
> > >
> > > Thank you for your kind reply! We will revise our paper based on your constructive comments.
> > >
> > > Regarding your point, it seems unlikely that we can achieve $O(\sqrt{𝑇})$ dynamic regret. In the worst case (i.e., $V_T=O(T)$), dynamic regret is expected to be linear in $𝑇$. We will add further clarifications to the paper.
> > >
> > > Best
> > >
> > > Authors

---

### Author Rebuttal · Authors · 2024-08-03

According to the suggestions of **Reviewers fFaA** and **PVQP**, we will reformulate our theorems to explicitly incorporate $\alpha$ and $\beta$. In this way, we obtain the regret bounds below, treating $\alpha$ and $\beta$ as constants.

---

***Theorem 1***. Under Assumptions 1 and 2, and setting $\eta = 1/\sqrt{d\gamma}$, Algorithm  2 ensures
$$
\begin{equation}
    \mathbb{E}\left [  R_D^* \right ]
    \le O \left (\frac{(1+\alpha \sqrt \gamma+ \beta \gamma)T}{\sqrt{\gamma }}  + \gamma V_T \right) . \nonumber
\end{equation}
$$
If the value of $V_T$ is known, we set $\gamma = \min \left \\{\left\lfloor (\frac{T}{V_T})^\frac{2}{3}\right\rfloor, T \right \\}$, then we have
$$
\begin{align}
    \mathbb{E}\left [ R_D^* \right ] &\le O((1+ \alpha \sqrt T+\beta T )T^\frac{2}{3}(V_T+1)^\frac{1}{3} ).\nonumber
\end{align}
$$
***Theorem 2.*** Let $\mathcal{H}=\left\\{\gamma_i = 2^i \mid  i=1,\cdots N\right\\} $ where $ N = \left\lfloor \log_{2}{T}\right\rfloor $, and $\rho=\frac{1}{dDL}\sqrt{\frac{8\ln{N}}{T}}$. Under Assumptions 1 and 2, Algorithm 3 ensures
$$
\begin{align}
\mathbb{E}\left [ R_D^*\right ]
\le O((1+ \alpha \sqrt T+\beta T )T^\frac{2}{3}(V_T+1)^\frac{1}{3} ).\nonumber
\end{align}
$$

***Theorem 3.*** Under Assumptions 1 and 2, Algorithm 4 ensures
$$
\begin{align}
    \mathbb{E}\left [ R_A(T,\tau )\right ]\le  O(\sqrt{\tau \log{T}} +\alpha \tau+ \beta \tau^{\frac{3}{2}} ).\nonumber
\end{align}
$$


***Theorem 4.*** Under Assumptions 1 and 2, Algorithm 4 ensures
$$
\begin{align}
    \mathbb{E}\left [ R_D^*\right ]
    \le \tilde{O}((1+ \alpha \sqrt T+\beta T )T^\frac{2}{3}(V_T+1)^\frac{1}{3}).\nonumber
\end{align}
$$

---

To simplify these theorems, we follow the seminal work of Suggala and Netrapalli [2020, Page 4] and also consider the case where $\alpha = O(1/\sqrt{T})$ and $\beta = O(1/T)$, thereby obtaining the same regret bounds as presented in our submitted manuscript.

---

### Decision · Program_Chairs · 2024-09-25

**Decision:**

Accept (poster)

**Comment:**

This paper provides an algorithm that, given an offline optimization oracle, can achieve optimal dynamic regret guarantees for non-convex losses.

In review, it appears that this paper's key contribution is to combine the recent results for obtaining static regret in this setting with a number of existing techniques for converting static regret into dynamic regret. This paper has borderline reviews, mostly because there is some concern about the level of contribution of this process. However, overall the reviews are more on the positive side. The authors are encouraged to address the reviewer comments and clarify their technical contributions in the revision.